# Metabolic and neurobehavioral disturbances induced by purine recycling deficiency in *Drosophila*

Céline Petitgas[1,2], Laurent Seugnet[3], Amina Dulac[1], Giorgio Matassi[4,5], Ali Mteyrek[1], Rebecca Fima[1], Marion Strehaiano[1], Joana Dagorret[1], Baya Chérif-Zahar[1], Sandrine Marie[6], Irène Ceballos-Picot[2], Serge Birman[1]*

[1]Genes Circuits Rhythms and Neuropathology, Brain Plasticity Unit, CNRS, ESPCI Paris, PSL Research University, Paris, France; [2]Metabolomic and Proteomic Biochemistry Laboratory, Necker-Enfants Malades Hospital and Paris Cité University, Paris, France; [3]Integrated Physiology of the Brain Arousal Systems (WAKING), Lyon Neuroscience Research Centre, INSERM/CNRS/UCBL1, Bron, France; [4]Dipartimento di Scienze Agroalimentari, Ambientali e Animali, University of Udine, Udine, Italy; [5]UMR "Ecology and Dynamics of Anthropogenic Systems" (EDYSAN), CNRS, Université de Picardie Jules Verne, Amiens, France; [6]Laboratory of Metabolic Diseases, Cliniques Universitaires Saint-Luc, Université catholique de Louvain, Brussels, Belgium

*For correspondence:
serge.birman@espci.fr

Competing interest: The authors declare that no competing interests exist.

**Abstract** Adenine phosphoribosyltransferase (APRT) and hypoxanthine-guanine phosphoribosyltransferase (HGPRT) are two structurally related enzymes involved in purine recycling in humans. Inherited mutations that suppress HGPRT activity are associated with Lesch–Nyhan disease (LND), a rare X-linked metabolic and neurological disorder in children, characterized by hyperuricemia, dystonia, and compulsive self-injury. To date, no treatment is available for these neurological defects and no animal model recapitulates all symptoms of LND patients. Here, we studied LND-related mechanisms in the fruit fly. By combining enzymatic assays and phylogenetic analysis, we confirm that no HGPRT activity is expressed in *Drosophila melanogaster*, making the APRT homolog (Aprt) the only purine-recycling enzyme in this organism. Whereas APRT deficiency does not trigger neurological defects in humans, we observed that *Drosophila Aprt* mutants show both metabolic and neurobehavioral disturbances, including increased uric acid levels, locomotor impairments, sleep alterations, seizure-like behavior, reduced lifespan, and reduction of adenosine signaling and content. Locomotor defects could be rescued by Aprt re-expression in neurons and reproduced by knocking down *Aprt* selectively in the protocerebral anterior medial (PAM) dopaminergic neurons, the mushroom bodies, or glia subsets. Ingestion of allopurinol rescued uric acid levels in *Aprt*-deficient mutants but not neurological defects, as is the case in LND patients, while feeding adenosine or $N^6$-methyladenosine (m⁶A) during development fully rescued the epileptic behavior. Intriguingly, pan-neuronal expression of an LND-associated mutant form of human HGPRT (I42T), but not the wild-type enzyme, resulted in early locomotor defects and seizure in flies, similar to *Aprt* deficiency. Overall, our results suggest that *Drosophila* could be used in different ways to better understand LND and seek a cure for this dramatic disease.

## eLife assessment

The article looks at how dysregulated purine metabolism in mutants for the *Aprt* gene impacts survival, motor, and sleep behavior in the fruit fly. Interestingly, although several deficits arise from

dopaminergic neurons, dopamine levels are increased in *Aprt* mutants. Instead, the biochemical change responsible for *Aprt* mutant neurobehavioral phenotypes appears to be a reduction in levels of adenosine. This **valuable** study suggests that *Drosophila Aprt* mutants may serve as a model for understanding Lesch–Nyhan disease (LND), caused by mutations in the human HPRT1 gene, and may also potentially serve as a model to screen for drugs for the neurobehavioral deficits observed in LND. The strength of evidence is **solid**.

## Introduction

The purine salvage pathway is an essential component of cellular metabolism that allows the recovery of free purine bases derived from the diet or from the degradation of nucleic acids and nucleotides, thus avoiding the energy cost of de novo purine biosynthesis (*Nyhan, 2014*). Energy-intensive tissues, such as cardiac muscle and brain cells, extensively use this pathway to maintain their purine levels (*Ipata, 2011*; *Johnson et al., 2019*). The two main recycling enzymes involved in the salvage pathway in mammals are hypoxanthine-guanine phosphoribosyltransferase (HGPRT), which converts hypoxanthine and guanine into IMP and GMP, respectively, and adenine phosphoribosyltransferase (APRT), which converts adenine into AMP.

APRT and HGPRT deficiencies induce very different disorders in humans. Loss of APRT seems to have only metabolic consequences, leading to the formation of 2,8-dihydroxyadenine crystals in kidney, which can be fatal but is readily prevented by allopurinol treatment (*Bollée et al., 2012*; *Harambat et al., 2012*). In contrast, highly inactivating mutations in HGPRT trigger Lesch–Nyhan disease (LND), a rare neurometabolic X-linked recessive disorder with dramatic consequences for child neurodevelopment (*Lesch and Nyhan, 1964*; *Seegmiller et al., 1967*). The metabolic consequence of HGPRT deficiency is an overproduction of uric acid in the blood (hyperuricemia) that can lead to gout and tophi, or nephrolithiasis (*Sass et al., 1965*; *Kelley et al., 1967*). Affected children also develop severe neurological impairments, such as dystonia, choreoathetosis, spasticity, and a dramatic compulsive self-injurious behavior (*Nyhan, 1997*; *Jinnah et al., 2006*; *Torres et al., 2007a*; *Schretlen et al., 2005*; *Madeo et al., 2019*). They have a developmental delay from 3 to 6 months after birth, and most of them never walk or even sit without support. Xanthine oxidase inhibitors, such as febuxostat or allopurinol, are given to patients after diagnosis to decrease their uric acid levels and prevent the formation of urate crystals in kidney, which can lead to renal failure (*Kelley et al., 1967*; *Torres et al., 2007a*; *Lahaye et al., 2014*). However, there is as yet no treatment to alleviate the neurological symptoms of LND (*Torres and Puig, 2007b*; *Jinnah et al., 2010*; *Madeo et al., 2019*).

To date, the causes of neurobehavioral troubles in LND are still not elucidated (*Jinnah et al., 2010*; *Bell et al., 2016*). The most favored hypothesis is a dysfunction of the brain's basal ganglia, and particularly of its dopaminergic pathways (*Baumeister and Frye, 1985*; *Visser et al., 2000*; *Nyhan, 2000*; *Saito and Takashima, 2000*; *Egami et al., 2007*). Indeed, analyses revealed a marked loss of dopamine (DA) (*Lloyd et al., 1981*; *Ernst et al., 1996*) and DA transporters (*Wong et al., 1996*) in the basal ganglia of LND patients. DA deficits have also been reported in HGPRT knockout rodents, but without motor or behavioral defects (*Finger et al., 1988*; *Dunnett et al., 1989*; *Jinnah et al., 1993*; *Jinnah et al., 1994*; *Meek et al., 2016*). Recent studies reported that HGPRT deficiency disrupts proliferation and migration of developing midbrain DA neurons in mouse embryos, arguing for a neurodevelopmental syndrome (*Witteveen et al., 2022*). This could result from ATP depletion and impaired energy metabolism (*Bell et al., 2021*) or an overactivation of de novo purine synthesis, leading to the accumulation of potentially toxic intermediates of this pathway (*López, 2008*; *López et al., 2020*). Pharmacological models have also been developed by injecting the neurotoxin 6-hydroxydopamine into neonatally rat brains, which induced a self-mutilation behavior in response to DA agonist administration in adulthood. However, these models are of limited utility as they do not reproduce the basic genetic impairment of LND (*Breese et al., 1990*; *Knapp and Breese, 2016*; *Bell et al., 2016*).

New animal models are therefore needed to study LND pathogenesis and find efficient therapeutic molecules. The fruit fly *Drosophila melanogaster* presents many advantages for translational studies and drug discovery (*Fernández-Hernández et al., 2016*; *Perrimon et al., 2016*; *Papanikolopoulou et al., 2019*). Although the importance of this invertebrate model for studying rare human genetic diseases is now recognized (*Oriel and Lasko, 2018*), a *Drosophila* LND model has not yet been developed to our knowledge. This is probably due to the fact that no HGPRT activity has been detected in

this organism (*Miller and Collins, 1973*; *Becker, 1974a*; *Becker, 1974b*). However, an ortholog of APRT is expressed in *Drosophila* (*Johnson et al., 1987*), encoded by the *Aprt* gene. It is therefore possible that part of the functions of human HGPRT, and in particular those essential for nervous system development and neurophysiology, are endorsed by Aprt in *Drosophila*.

Here, we studied the effects of *Aprt* deficiency on purine metabolism, lifespan, and various adult fly behaviors, including spontaneous and startle-induced locomotion, sleep, and seizure-like bang sensitivity (BS). We show that Aprt is required during development and in adult stage for many aspects of *Drosophila* life, and that its activity is of particular importance in subpopulations of brain dopaminergic neurons and glial cells. Lack of Aprt appears to decrease adenosinergic signaling and induces both metabolic and neurobehavioral symptoms in flies, as is the case with HGPRT in humans. We also find that expression of an LND-associated mutant form of HGPRT, but not the wild-type enzyme, in *Drosophila* neurons, induced neurobehavioral impairments similar to those of *Aprt*-deficient flies. Such a potential toxic gain-of-function effect of mutated HGPRT had not yet been demonstrated in an animal model .

**Table 1.** Aprt activity in wild-type and Aprt-deficient flies.

| Genotypes | Sex | Aprt activity (nmol/min/mg prot) |
|---|---|---|
| Wild type | Males | 1.32 ± 0.17 |
| | Females | 2.77 ± 0.27 |
| *Aprt⁵/Aprt⁵* | Males and females | 0.04 ± 0.02 |
| *Aprt⁵/Df(3L)ED4284* | Males | 0.02 ± 0.01 |
| *da/+* | Males | 2.78 ± 0.41 |
| *da>Aprt^RNAi* | Males | 0.10 ± 0.01 |
| *Aprt^RNAi/+* | Males | 2.16 ± 0.37 |

The online version of this article includes the following source data for table 1:

**Source data 1.** Source data for *Table 1*.

## Results

### Evolution of HGPRT proteins

The pathways of purine anabolism/catabolism and recycling have been closely conserved between *Drosophila melanogaster* and humans (*Figure 1—figure supplement 1*): all genes related to purine metabolism have homologs in both species, except for the human *HPRT1* gene encoding HGPRT (step 13 in *Figure 1—figure supplement 1*), which has no ortholog in flies, and the lack of urate oxidase in humans (step 20). In accordance with pioneering reports from about 50 years ago (*Miller and Collins, 1973*; *Becker, 1974a*; *Becker, 1974b*), we confirmed that no HGPRT enzymatic activity can be detected in extracts of wild-type *D. melanogaster* (see below Table 2), using either hypoxanthine or guanine as substrate in the reaction medium. This intriguing observation prompted us to carry out a more precise analysis of the evolution of HGPRT.

HGPRT proteins are ancient, for they are present in both bacteria and archaea. However, the analysis of the phyletic distribution of HGPRT proteins revealed their striking rareness in insecta. This conclusion is based on PSI-Blast sequence similarity searches on the NCBI Insecta database (taxid: 6960, 50557). Phylogenetic analysis showed that the only 11 HGPRT proteins found in Insecta cluster mainly with bacteria, but also with fungi, apicomplexa, and acari (*Figure 1—figure supplement 2*, red font, see legend for details). These and further evidence support the acquisition of HGPRT in a few insecta species by horizontal gene transfer (G. Matassi, unpublished observations). In particular, HGPRT has no homolog in Drosophilidae, with the potential exception of a single species, *Drosophila immigrans*, in which our most recent PSI-BLAST analysis detected one hit (accession KAH8256851.1, annotated as hypothetical protein). Yet this sequence is 100% identical to the HGPRT of the Gammaproteobacterium *Serratia marcescens*. A phylogenetic analysis showed that the *D. immigrans* HGPRT clusters with the *Serratia* genus (*Figure 1—figure supplement 3*), suggesting either a contamination of the sequenced sample or a very recent horizontal gene transfer event. The second scenario is more likely since the corresponding nucleotide sequences differ by five synonymous substitutions (out of 534 positions). We also carried out structural similarity searches against the RCSB Protein Data Bank repository using the human HGPRT structure as query (PDB identifiers: 5HIA or 1Z7G). This analysis did not identify any protein with a divergent sequence and relevant similarity with HGPRT 3D structure in *D. melanogaster*, consistent with the lack of HGPRT enzymatic activity in this organism.

### *Drosophila* lacking Aprt activity have a shortened lifespan

Both the phylogeny and enzymatic assays therefore strongly suggest that Aprt (*Figure 1—figure supplement 1*, step 12) is the only recycling enzyme of the purine salvage pathway in *Drosophila*. To assess the importance of purine recycling in brain development and function in this organism, we analyzed the phenotypes induced by a deficiency in Aprt. The *Aprt*[5] mutant was originally generated by chemical mutagenesis followed by selection of flies resistant to purine toxicity (*Johnson and Friedman, 1981*; *Johnson and Friedman, 1983*). Enzymatic assays confirmed a strong reduction in Aprt activity in extracts of heterozygous *Aprt*[5]/+ mutants and its absence in homozygous and hemizygous (*Aprt*[5]/*Df(3L)ED4284*) mutants (*Figure 1—figure supplement 4* and *Table 1*). Sequencing of the *Aprt*[5] cDNA (*Figure 1—figure supplement 5A*) indicated that the *Aprt*[5] mRNA codes for a protein with several amino acid changes compared to *D. melanogaster* wild-type Aprt, modifying in particular three amino acid residues that have been conserved in the Aprt sequences from *Drosophila* to humans (*Figure 1—figure supplement 5B*). These mutations are likely to be responsible for the loss of enzymatic activity. The homozygous *Aprt*[5] mutants are considered viable because they develop and reproduce normally. However, we observed that these mutants have a significantly reduced longevity, their median lifespan being only 38 d against 50 d for wild-type flies (p<0.001) (*Figure 1A*).

### Uric acid levels are increased in *Aprt*[5] mutants and rescued by allopurinol

In humans, one of the consequences of HGPRT deficiency on metabolism is the overproduction of uric acid (*Harkness et al., 1988*; *Fu et al., 2015*). We assayed the levels of purine metabolites by HPLC and found that the level of uric acid was substantially increased by 37.7% on average in *Drosophila Aprt*[5] mutant heads (p<0.01) (*Figure 1B*). We then tried to rescue uric acid content in flies by providing allopurinol in the diet, as is usually done for LND patients. Allopurinol is a hypoxanthine analog and a competitive inhibitor of xanthine oxidase, an enzyme that catalyzes the oxidation of xanthine into uric acid (*Figure 1—figure supplement 1*, step 19). Remarkably, the administration of 100 µg/ml allopurinol during 5 d decreased uric acid levels to a normal concentration range in *Aprt*[5] mutant heads (*Figure 1B*). Therefore, quite similarly to HGPRT deficit in humans, the lack of Aprt activity in flies increases uric acid levels and this metabolic disturbance can be prevented by allopurinol.

### Aprt deficiency decreases motricity in young flies

LND patients present dramatic motor disorders that prevent them for walking at an early age. To examine whether a deficiency in the purine salvage pathway can induce motor disturbance in young flies, we monitored the performance of *Aprt*-null mutants in startle-induced negative geotaxis (SING), a widely used paradigm to assess climbing performance and locomotor reactivity (*Feany and Bender, 2000*; *Friggi-Grelin et al., 2003*; *Riemensperger et al., 2013*; *Sun et al., 2018*). Strikingly, *Aprt*[5] mutant flies showed a very early SING defect starting from 1 day after eclosion (d a.E.) (performance index [PI] = 0.73 vs 0.98 for wild-type flies, p<0.001) that was more pronounced at 8 d a.E. (PI = 0.51 vs 0.96 for the wild-type flies, p<0.001). The fly locomotor performance did not further decline afterwards until 30 d a.E. (*Figure 1C*). *Df(3L)ED4284* and *Df(3L)BSC365* are two partially overlapping small genomic deficiencies that uncovers *Aprt* and several neighbor genes. Hemizygous *Aprt*[5]/*Df(3L)ED4284* or *Aprt*[5]/*Df(3L)BSC365* flies also displayed SING defects at 10 d a.E. (PI = 0.71 and 0.68, respectively, compared to 0.97 for wild-type flies, p<0.01) (*Figure 1—figure supplement 6*). In contrast to its beneficial effect on uric acid levels, we observed that allopurinol treatment did not improve the locomotor ability of *Aprt*[5] mutant flies, either administered 5 d before the test or throughout the development (*Figure 1D and E*). This is comparable to the lack of effect of this drug against neurological defects in LND patients.

To confirm the effect of *Aprt* deficiency on the SING behavior, we used an *UAS-Aprt*[RNAi] line that reduced Aprt activity by more than 95% when expressed in all cells with the *da-Gal4* driver (*Table 1*). These *Aprt* knock-down flies also showed a strong SING defect at 10 d a.E. (PI = 0.62 against 0.97 and 0.85 for the driver and *UAS-Aprt*[RNAi] alone controls, respectively, p<0.001), like the *Aprt*[5] mutant (*Figure 1F*). Next, we used *tub-Gal80*[ts], which inhibits Gal4 activity at permissive temperature (*McGuire et al., 2003*), to prevent Aprt knockdown before the adult stage. *Tub-Gal80*[ts]; *da-Gal4>Aprt*[RNAi] flies raised at permissive temperature (18°C) did not show any locomotor impairment (*Figure 1G*, white bars). However, after being transferred for 3 d (between 7 and 10 d a.E.) at a restrictive temperature

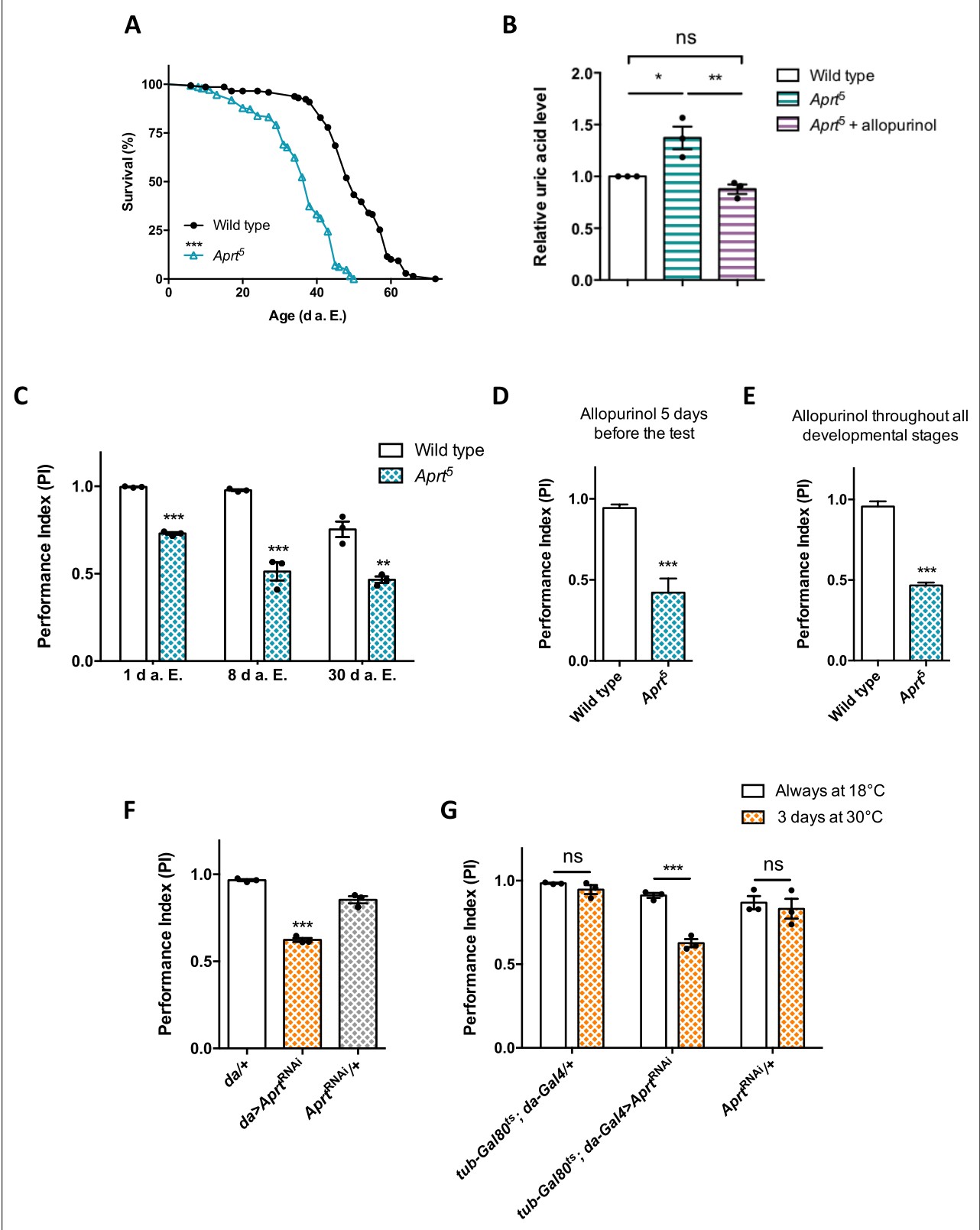

**Figure 1.** Aprt deficiency shortens lifespan and induces metabolic and neurobehavioral defects. (**A**) *Aprt*[5] mutant flies have a reduced lifespan compared to wild-type flies (median lifespan: 38 and 50 d, respectively). Three independent experiments were performed on 150 males per genotype with similar results and a representative experiment is shown. Log-rank test (***p<0.001). (**B**) HPLC profiles on head extracts revealed an increase in uric acid levels in *Aprt*[5] mutant flies. Administration of 100 μg/ml allopurinol for 5 d before the test rescued uric acid levels. Mean of three independent

*Figure 1 continued on next page*

*Figure 1 continued*

experiments performed on 40 flies per genotype. One-way ANOVA with Tukey's *post hoc* test for multiple comparisons (*p<0.05; **p<0.01; ns: not significant). (**C–E**) Effect on climbing ability. (**C**) *Aprt⁵* mutants shows early defects in the startle-induced negative geotaxis (SING) paradigm that monitors locomotor reactivity and climbing performance. This deficit was already obvious at 1 day after eclosion (d a.E.) and further decreased up to 8 d a.E., after which it did not change significantly up to 30 d a.E. Mean of three independent experiments performed on 50 flies per genotype. Unpaired Student's *t*-test (**p<0.01; ***p<0.001). (**D, E**) Administration of allopurinol does not rescue the motricity defects of *Aprt*-deficient mutants. Feeding the *Aprt*5 mutants with allopurinol (100 μg/ml) either in adults 5 d before the test (**D**) or throughout all developmental stages (**E**) did not alter the defects observed in SING behavior. Results of one experiment performed on 50 flies per genotype at 10 d a. E. Unpaired Student's *t*-test (***p<0.001). (**F**) Downregulating *Aprt* by RNAi in all cells (*da>Aprt^RNAi*) also led to an early impairment in climbing responses in the SING assay at 10 d a.E. compared to the driver (*da/+*) and effector (*Aprt^RNAi/+*) only controls. Mean of three independent experiments performed on 50 flies per genotype. One-way ANOVA with Tukey's *post hoc* test for multiple comparisons (***p<0.001). (**G**) Adult-specific inactivation of *Aprt* (*tub-Gal80^ts; da-Gal4>Aprt^RNAi*) decreased startle-induced climbing abilities in the SING paradigm, suggesting that the locomotor impairment induced by Aprt deficiency is not caused by a developmental effect. Flies were raised at permissive temperature (18°C) in which Gal80^ts suppressed Gal4-controlled *Aprt^RNAi* expression and were shifted from 18 to 30°C for 3 d before the test (between 7 and 10 d a.E.) to activate transgene expression. Mean of three independent experiments performed on 50 flies per genotype. Two-way ANOVA with Sidak's *post hoc* test for multiple comparisons (***p<0.001; ns: not significant).

The online version of this article includes the following source data and figure supplement(s) for figure 1:

**Source data 1.** Source data for *Figure 1A–G*.

**Figure supplement 1.** Comparison of purine metabolism pathways in *Drosophila* and humans.

**Figure supplement 2.** Urooted maximum likelihood phylogeny of hypoxanthine-guanine phosphoribosyltransferase (HGPRT) proteins (189 taxa, 130 sites).

**Figure supplement 3.** Urooted maximum likelihood phylogeny of HPRT proteins (20 taxa, 177 sites).

**Figure supplement 4.** Lack of Aprt enzymatic activity in the *Aprt⁵* mutant.

**Figure supplement 4—source data 1.** Source data for *Figure 1—figure supplement 4*.

**Figure supplement 5.** Alignment of wild-type and mutant *Aprt* cDNAs and predicted protein sequences.

**Figure supplement 6.** Startle-induced negative geotaxis (SING) behavior of hemizygous *Aprt* mutant flies.

**Figure supplement 6—source data 1.** Source data for *Figure 1—figure supplement 6*.

(30°C) that inactivates Gal80, the same flies demonstrated a similar SING defect as *Aprt⁵* mutants (PI = 0.63 compared to 0.94 and 0.83 for the driver and RNAi alone controls, respectively, p<0.001) (*Figure 1G*, yellow bars with dots). This demonstrates that *Aprt* inactivation at the adult stage only is sufficient to alter SING performance in *Drosophila*, strongly suggesting that this genotype does not result from developmental defects.

## Cell specificity of Aprt requirement for startle-induced locomotion

We then tried to identify the neural cells in which Aprt is required to ensure a normal locomotor reactivity in young flies by expressing *Aprt^RNAi* with selective Gal4 drivers. Expression in all neurons with *elav-Gal4* led to decreased locomotor performance in the SING test (PI = 0.68 at 10 d a.E., vs 0.90 and 0.86 for the driver and RNAi controls, respectively, p<0.05) (*Figure 2A*), which was comparable to the effects observed with the *Aprt⁵* mutant or after ubiquitous expression of the RNAi. To confirm the role of neuronal Aprt in locomotor control, we generated a *UAS-Aprt* line, which allowed for a substantial increase in Aprt expression and activity (*Figure 2—figure supplement 1*). We then found that re-expressing *Drosophila* Aprt selectively in neurons in *Aprt⁵* background partially rescued the SING phenotype of the null mutant (PI = 0.70 vs 0.52 for *driver* and *UAS-Aprt* controls in *Aprt⁵* background, p<0.05) (*Figure 2B*).

Furthermore, *Aprt* knockdown in all glial cells with *repo-Gal4*, or in sub-populations of glial cells that express the glutamate transporter Eaat1 with *Eaat1-Gal4*, which includes astrocyte-like glia, cortex glia, and some subperineurial glia (*Rival et al., 2004*; *Mazaud et al., 2019*), also led to a lower SING performance of 10-day-old flies (PI = 0.72 for *repo-Gal4* vs 0.91 for both controls, p<0.05, and PI = 0.56 for *Eaat1-Gal4* vs 0.77, p<0.05, and 0.88, p<0.01, for the driver and RNAi controls, respectively) (*Figure 2C and D*). In contrast, *MZ0709-Gal4* (*Doherty et al., 2009*) and *NP6520-Gal4* (*Awasaki et al., 2008*) that selectively target the ensheathing glia did not induce any significant locomotor defects when used to express the *Aprt* RNAi (*Figure 2—figure supplement 2*). Noticeably, re-expressing Aprt with *Eaat1-Gal4* in the *Aprt⁵* background did not rescue the SING phenotype (*Figure 2E*), in contrast to the positive effect of neuronal re-expression (*Figure 2B*). Overall this

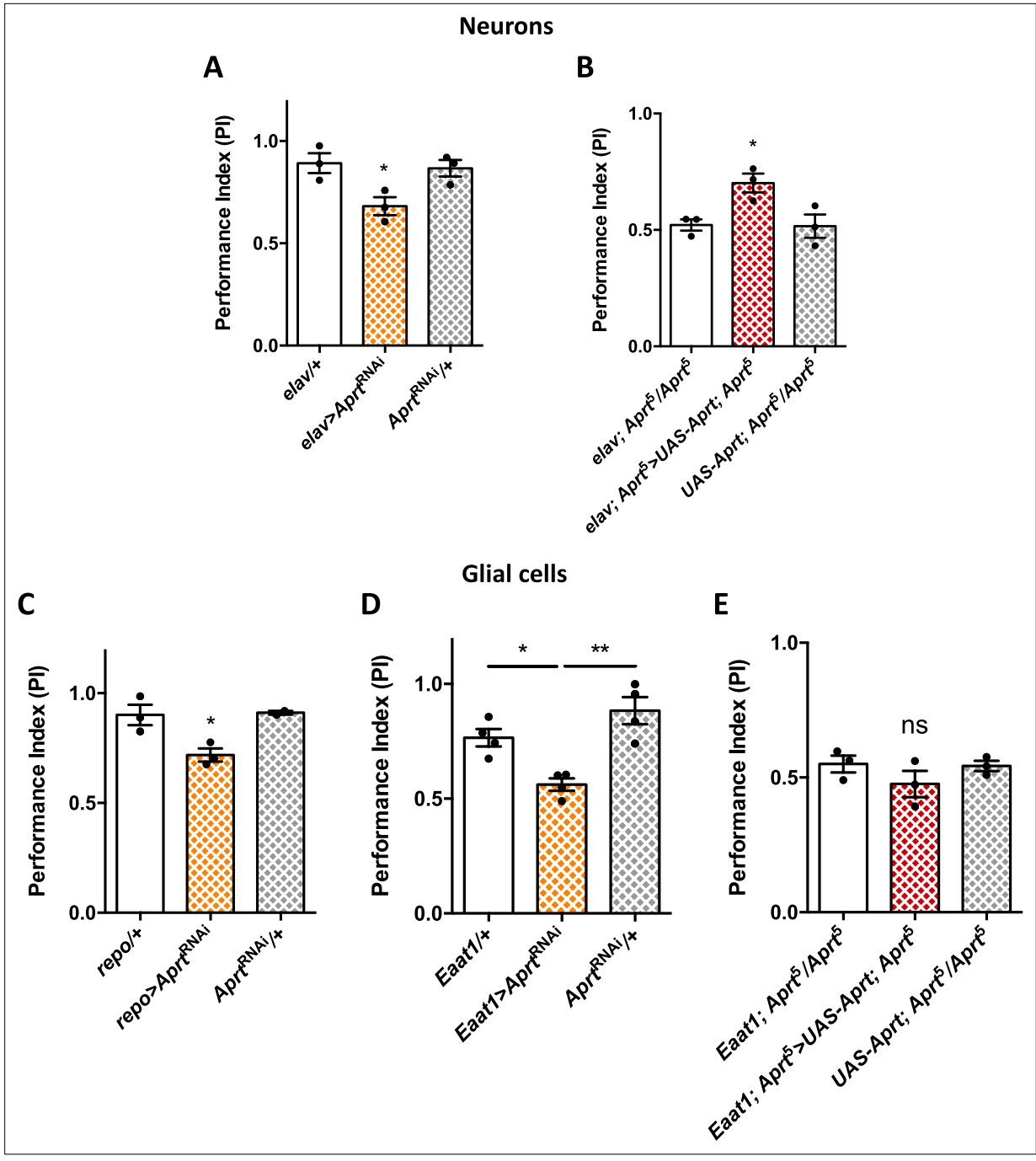

**Figure 2.** *Aprt* knockdown in neurons or glial cells disrupts startle-induced locomotion in *Drosophila*. (**A**) *Aprt*$^{RNAi}$ expression in all neurons with *elav-Gal4* decreased startle-induced negative geotaxis (SING) performance in 10-day-old flies. (**B**) Pan-neuronal expression of *Drosophila Aprt* with the UAS-Gal4 system partially rescued the locomotor response of *Aprt*$^5$ mutants. (**C, D**) Downregulation of *Aprt* expression in all glia with *repo-Gal4* (**C**) or in glial cell that express the glutamate transporter Eaat1 with *Eaat1-Gal4* (**D**) also altered SING performances. (**E**) *Aprt* re-expression in glial cells with *Eaat1-Gal4* driver did not rescue the climbing response of *Aprt*$^5$ mutants. Results of three or four independent experiments performed on 50 flies per genotype at 10 days after eclosion (d a.E.). One-way ANOVA with Tukey's *post hoc* test for multiple comparisons (*$p < 0.05$; **$p < 0.01$; ns: not significant).

The online version of this article includes the following source data and figure supplement(s) for figure 2:

**Source data 1.** Source data for *Figure 2A–E*.

**Figure supplement 1.** Transgenic expression of *Drosophila* Aprt.

**Figure supplement 1—source data 1.** Source data for *Figure 2—figure supplement 1*.

**Figure supplement 2.** Downregulation of *Aprt* expression in the ensheathing glia does not alter locomotor performances.

**Figure supplement 2—source data 1.** Source data for *Figure 2—figure supplement 2*.

suggests that Aprt is required both in neurons and glia subsets to ensure a normal SING performance in young flies, and that neuronal but not glial *Aprt* re-expression is sufficient to restore a partially functional locomotor behavior.

To identify the neuronal subpopulations in which Aprt is required to ensure proper locomotor responses in young flies, we first tested dopaminergic drivers since we previously showed that DA neurons play an important role in the control of the SING behavior (*Riemensperger et al., 2011*; *Riemensperger et al., 2013*; *Sun et al., 2018*). *Aprt* knockdown targeted to these neurons with *TH-Gal4* did not induce significant impairments (*Figure 3A*). This driver expresses in all brain DA neurons except a large part of the protocerebral anterior medial (PAM) cluster (*Friggi-Grelin et al., 2003*; *Mao and Davis, 2009*; *Pech et al., 2013*). In contrast, downregulating *Aprt* with the *TH-Gal4, R58E02-Gal4* double driver, which labels all DA neurons including the PAM cluster (*Sun et al., 2018*), induced a quite similar locomotor defect as did pan-neuronal *Aprt* knockdown (PI = 0.72 vs 0.96 and 0.93 for the driver and RNAi controls, respectively, p<0.001) (*Figure 3B*). Besides, downregulating *Aprt* in a majority of the serotonergic neurons with *TRH-Gal4* (*Cassar et al., 2015*) did not induce a SING defect (*Figure 3C*).

These results suggest that some DA neurons in the PAM cluster are specifically involved in the locomotor impairments induced by *Aprt* deficiency. It has been previously shown in our laboratory that inhibiting DA synthesis in a subset of 15 PAM DA neurons cluster was able to alter markedly SING performance in aging flies (*Riemensperger et al., 2013*). We and others also reported that the degeneration or loss of PAM DA neurons is involved in the SING defects observed in several *Drosophila* models of Parkinson's disease (*Riemensperger et al., 2013*; *Bou Dib et al., 2014*; *Tas et al., 2018*; *Pütz et al., 2021*). Here we found that expressing *Aprt*[RNAi] in the PAM cluster with *R58E02-Gal4* reproduced the same motor disturbance as pan-neuronal expression (PI = 0.74 vs 0.96, p<0.001, and 0.85, p<0.05, for the driver and RNAi controls, respectively) (*Figure 3D*), and this result was confirmed by using two other PAM drivers (*NP6510-Gal4* – expressing only in 15 neurons – and *R76F05-Gal4*) (*Figure 3E and F*). This strongly suggests that purine recycling deficiency compromises the correct functioning of these neurons, leading to a defective startle-induced locomotion.

Because PAM DA neurons innervate the horizontal lobes of the mushroom bodies (*Liu et al., 2012*; *Riemensperger et al., 2013*), and because this structure has been shown to be involved in the control of startle-induced climbing (*Riemensperger et al., 2013*; *Bou Dib et al., 2014*; *Sun et al., 2018*), we also inactivated *Aprt* in mushroom body neurons with *238Y-Gal4* that strongly expresses in that structure (*Aso et al., 2009*). Interestingly, this driver did induce a locomotor reactivity impairment (PI = 0.70 vs 0.89 for both controls, p<0.01) (*Figure 3G*), and the same result was observed with *VT30559-Gal4*, which is a very specific driver for all the mushroom body intrinsic neurons (*Plaçais et al., 2017*; *Figure 3H*). Overall, these results show that normal expression of the SING behavior depends on *Aprt* expression in the PAM and mushroom body neurons in *Drosophila*.

## Sleep disturbances induced by Aprt deficiency

Both the mushroom body and subpopulations of PAM DA neurons are known to be regulators of sleep in *Drosophila* (*Nall et al., 2016*; *Artiushin and Sehgal, 2017*). The fact that *Aprt* deficiency in some of these cells impaired locomotor regulation prompted us to monitor the spontaneous locomotion and the sleep pattern of *Aprt* mutants. Compared to controls, *Aprt*-deficient flies did not have an altered circadian activity profile (*Figure 4—figure supplement 1*), nor any difference in total spontaneous locomotor activity during the day, as quantified by the number of infrared beam cuts (*events*) per 30 min, but they showed a 26.2% increase in total activity during the night (p<0.001) (*Figure 4A*). As usual, a sleep bout was defined as 5 min or more of fly immobility (*Huber et al., 2004*), and we checked that wild-type and *Aprt* mutant flies that did not move for 5 min were indeed asleep because they were less sensitive to mild mechanical stimulation (*Figure 4—figure supplement 2*). We found that *Aprt*[5] mutants slept significantly less than wild-type flies and that it was the case both during day and night (*Figure 4B and C*). These mutants indeed showed a reduced walking speed (*Figure 4D*) and a smaller average sleep bout duration (*Figure 4E*), indicating a difficulty to maintain sleep. The reduced speed does not seem to be caused by a decreased energetic metabolism since we could not detect different ATP levels in head and thorax of *Aprt*[5] mutants compared to wild-type flies (*Figure 4—figure supplement 3*).

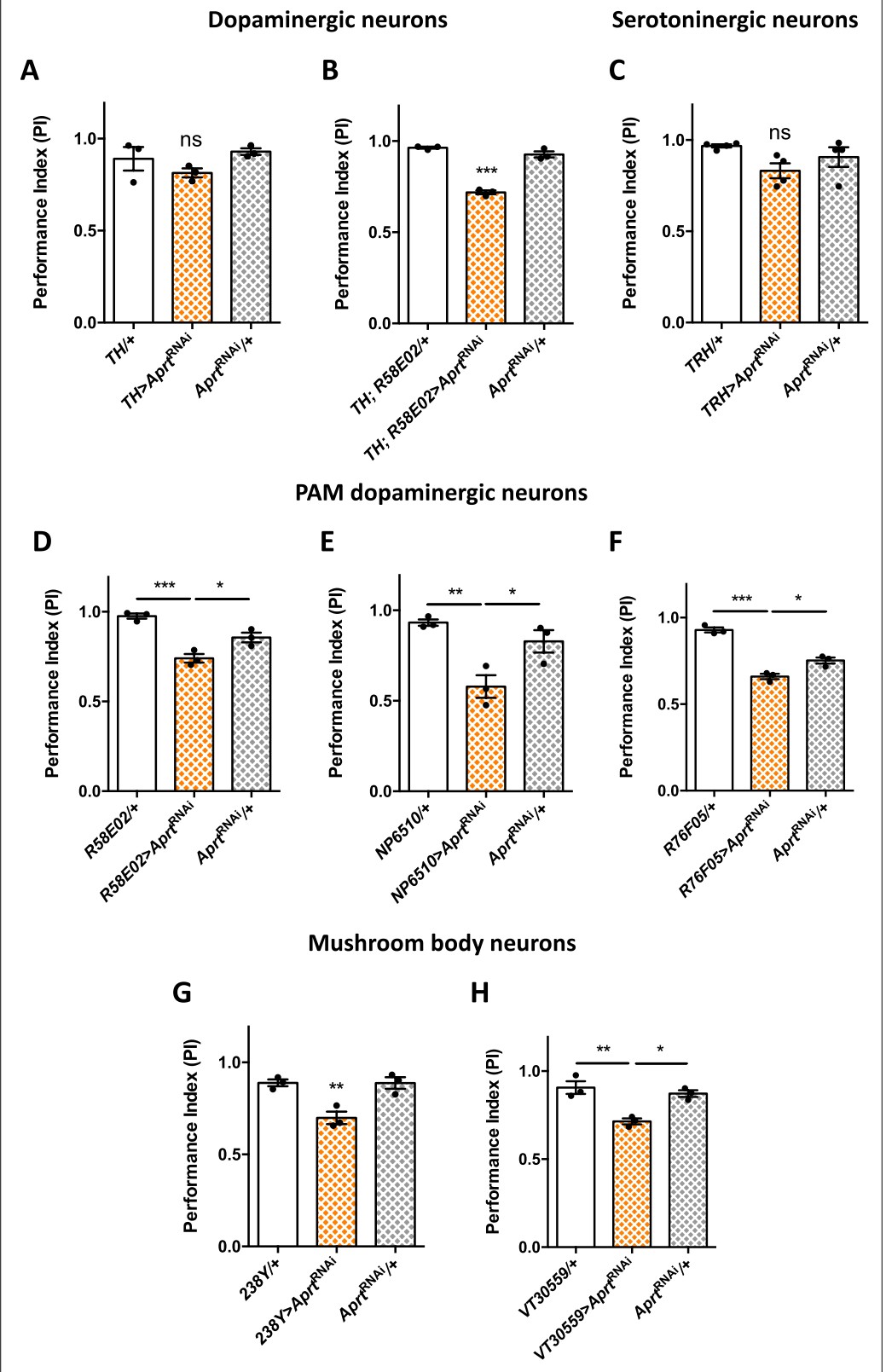

**Figure 3.** *Aprt* downregulation in dopamine (DA) neurons of the protocerebral anterior medial (PAM) cluster and in mushroom body neurons impairs startle-induced locomotion. (**A**) RNAi-mediated *Aprt* inactivation in brain DA neurons except a large part of the PAM cluster with the *TH-Gal4* driver did not lead to locomotor defects in the startle-induced negative geotaxis (SING) assay. (**B**) In contrast, *Aprt* knockdown in all dopaminergic neurons

*Figure 3 continued on next page*

*Figure 3 continued*

including the PAM cluster with the *TH-Gal4, R58E02-Gal4* double-driver led to a decrease in SING performance. (**C**) *Aprt* downregulation in serotonergic neurons with *TRH-Gal4* did not alter startle-induced climbing response of the flies. (**D–F**) *Aprt* knockdown selectively in DA neurons of the PAM cluster using either *R58E02-Gal4* (**D**), *NP6510-Gal4* (**E**), or *R76F05-Gal4* (**F**) significantly decreased climbing performance. (**G, H**) *Aprt* knockdown in all the mushroom body intrinsic neurons (Kenyon cells) with *238Y-Gal4* (**G**) or *VT30559-Gal4* (**H**) also led to a decrease in SING performance. Results of three or four independent experiments performed on 50 flies per genotype at 10 days after eclosion (d a.E.). One-way ANOVA with Tukey's *post hoc* test for multiple comparisons (*p<0.05; **p<0.01; ***p<0.001; ns: not significant).

The online version of this article includes the following source data for figure 3:

**Source data 1.** Source data for *Figure 3A–H*.

Interestingly, RNAi-mediated downregulation of Aprt selectively in neurons (*elav>Aprt*[RNAi] flies) also significantly decreased sleep duration during day and night (*Figure 4F*), whereas glial-only expression (*repo>Aprt*[RNAi]) had no effect (*Figure 4—figure supplement 4A*). Expressing the RNAi in both neurons and glia (*elav; repo>Aprt*[RNAi]) had similar effects as in neurons alone (*Figure 4—figure supplement 4B*). Quantification of total sleep in these experiments, and the total amount of day and night sleep, are shown in *Figure 4G* and *Figure 4—figure supplement 4C and D*, respectively. Overall, these results suggest that *Aprt* expression is selectively needed in neurons for a normal sleep pattern in *Drosophila*.

## Brain DA synthesis and levels are increased in *Aprt*-deficient flies

Since we found that induced locomotion and sleep, two behaviors controlled by DA neurons in *Drosophila*, were altered in *Aprt*-deficient flies, we expected brain DA levels to be reduced in these mutants, as is the case in the basal ganglia of LND patients. We therefore carried out comparative immunostaining for tyrosine hydroxylase (TH), the specific enzyme for DA synthesis (*Friggi-Grelin et al., 2003*; *Riemensperger et al., 2011*), on dissected adult brains from wild-type and *Aprt*[5] mutant flies. However, the global TH protein level appeared not to be decreased, but relatively increased by 17.5% in the *Aprt*[5] mutant brain (p<0.01), in particular around the mushroom bodies, a structure that receives dense dopaminergic projections (*Figure 5A and B*). Moreover, DA immunostaining carried out on whole-mount dissected brains revealed a similarly increased level of this neuromodulator in *Aprt*[5] flies, by 17.0% on average in the entire brain (p<0.01), but not specifically in the mushroom bodies or another part of the brain (*Figure 5C and D*). We also found that the transcript levels of *DTH1*, encoding the TH neuronal isoform in *Drosophila*, were increased in *Aprt*[5] mutants compared to wild-type flies (*Figure 5E*), and, conversely, decreased when Aprt was overexpressed ubiquitously (*Figure 5F*). Western blot experiments further indicated that DTH1 protein levels are increased in *Aprt*[5] compared to controls (*Figure 5G and H*). This indicates that disruption of the purine salvage pathway does not impede DA synthesis and levels in the *Drosophila* brain, which are instead slightly increased. We therefore searched for another neurotransmitter system that could be affected by *Aprt* deficiency.

## Interactions between Aprt and the adenosinergic system

Aprt catalyzes the conversion of adenine into AMP, and AMP breakdown by the enzyme 5'-nucleotidase produces adenosine, primarily in the extracellular space. Adenosine then acts as a widespread neuromodulator in the nervous system by binding to adenosine receptors. We therefore suspected that adenosine level could be reduced in the absence of Aprt activity, leading to alterations in adenosinergic neurotransmission. Indeed, we found a significant decrease in adenosine level either in whole flies (by 61.0% on average, p<0.01) or in heads (by 48.0%, p<0.05) in *Aprt*[5] mutants (*Figure 6A*). We then examined the consequences of this reduction on molecular components of the adenosinergic system, namely G protein-coupled adenosine receptors and equilibrative nucleoside transporters (ENTs), which carry out nucleobase and nucleoside uptake of into cells. Only one seven-transmembrane-domain adenosine receptor, AdoR, is present in *Drosophila*, which is very similar to mammalian adenosine receptors (*Dolezelova et al., 2007*; *Brody and Cravchik, 2000*), and three putative equilibrative nucleoside transporters (Ent1-3) have been identified (*Sankar et al., 2002*) but only one, Ent2, showed nucleoside transport activity (*Machado et al., 2007*). Interestingly, Ent2 has

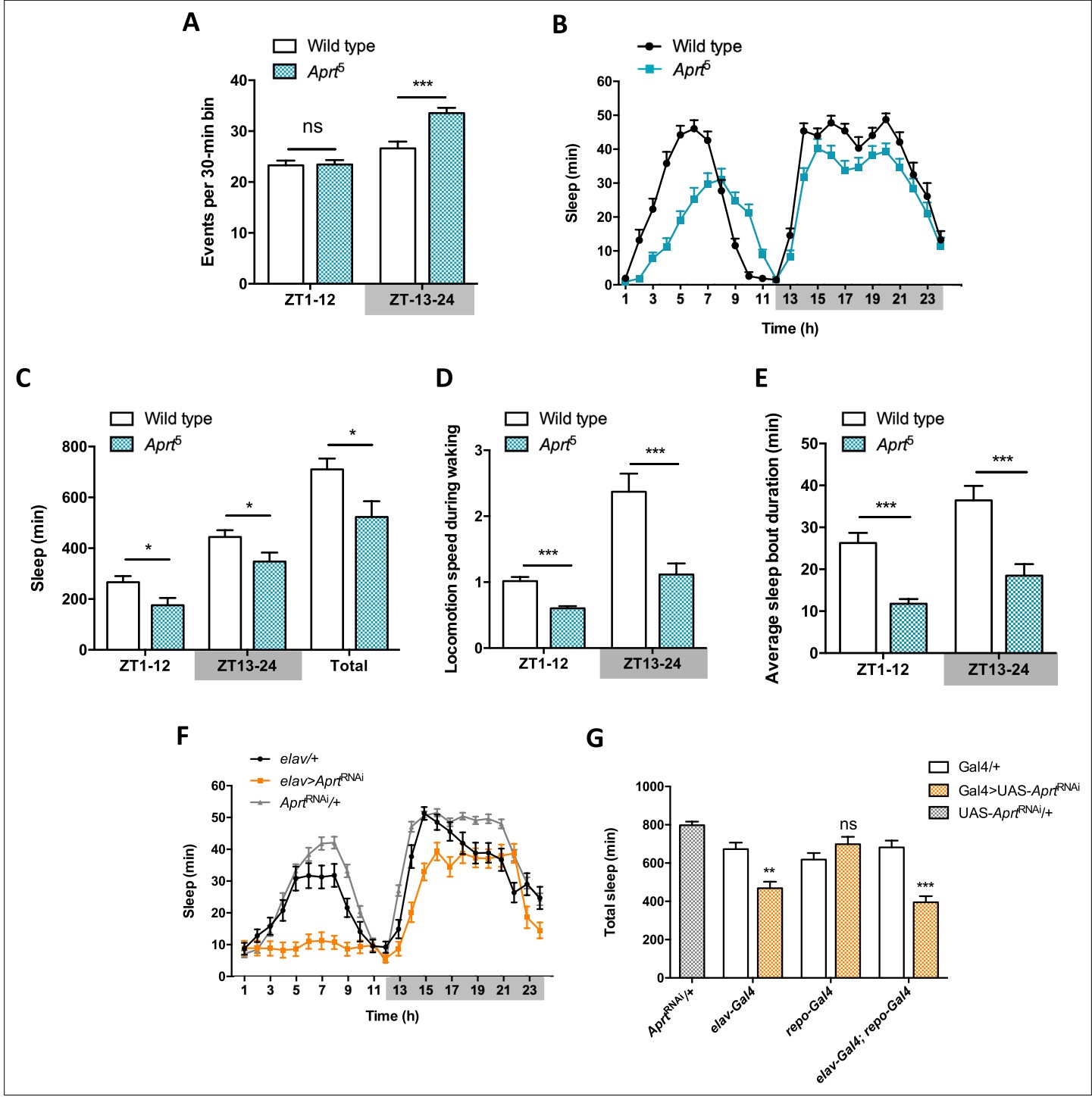

**Figure 4.** *Aprt*-deficient flies sleep less and walk slower than wild-type flies. (**A**) Quantification of total spontaneous locomotor activity during day and night over five light-dark (LD) cycles. *Aprt⁵* mutants show no difference in spontaneous locomotion with wild-type flies during the day but a higher activity at night. Three independent experiments were performed on 32 flies per genotype and mean ± SEM was plotted. Unpaired Student's *t*-test (***p<0.001; ns: not significant). (**B**) Sleep pattern during a typical 24 hr LD cycle showing that the total amount of sleep is smaller during day and night in *Aprt⁵* mutants compared to wild-type flies. (**C**) Quantification of day (ZT1-12), night (ZT13-24), and total sleep in *Aprt⁵* mutants. ZT, zeitgeber. (**D**) Locomotion speed during waking is reduced in *Aprt⁵* mutants. (**E**) The average sleep bout duration is also decreased, indicating that *Aprt⁵* mutants have a difficulty to maintain sleep. (**F**) Sleep pattern of *elav>Aprt*^RNAi flies, showing that knockdown of Aprt in all neurons led to sleep reduction during the night, similarly to the mutant condition, and an even more profound sleep defect during the day. (**G**) Quantification of total amount of sleep when *Aprt* was downregulated in all neurons (*elav-Gal4*), all glial cells (*repo-Gal4*), and both neurons and glial cells (*elav-Gal4; repo-Gal4*). Except for glia only,

*Figure 4 continued on next page*

*Figure 4 continued*

the resulting effect was a significant sleep reduction. For sleep and locomotion speed, means ± SEM were plotted. Unpaired Student's *t*-test (**C–E**) and one-way ANOVA with Tukey's *post hoc* test for multiple comparisons (**G**) (*p<0.05; **p<0.01; ***p<0.001; ns: not significant).

The online version of this article includes the following source data and figure supplement(s) for figure 4:

**Source data 1.** Source data for *Figure 4A–G*.

**Figure supplement 1.** Daily locomotor activity profiles of wild-type and *Aprt*[5] mutant flies.

**Figure supplement 2.** Sensitivity of resting flies to mild mechanical stimulation.

**Figure supplement 2—source data 1.** Source data for *Figure 4—figure supplement 2*.

**Figure supplement 3.** ATP levels are not altered in head and thorax of *Aprt*[5] flies compared to the wild-type.

**Figure supplement 3—source data 1.** Source data for *Figure 4—figure supplement 3*.

**Figure supplement 4.** Sleep patterns of flies with cell-specific *Aprt* deficiency.

**Figure supplement 4—source data 1.** Source data for *Figure 4—figure supplement 4*.

been shown to be enriched in the mushroom bodies and its transcript level to be highly elevated in *AdoR* mutant flies (*Knight et al., 2010*). In *Aprt*[5] background, we also observed a strong increase in *Ent2* mRNAs (2.3-fold higher than wild-type flies, p<0.001), but no noticeable effect on *AdoR* transcript level (*Figure 6B*). The increased expression of *Ent2* could correspond to a compensatory response to adenosine shortage in *Aprt*[5] mutants.

The AdoR receptor is highly expressed in adult fly heads and its ectopic overexpression leads to early larval lethality (*Dolezelova et al., 2007*). In contrast, a null mutant of this receptor, *AdoR*[KGex], in which the entire coding sequence is deleted, is fully viable (*Wu et al., 2009*). This enabled us to examine the consequences of a complete lack of AdoR on purine recycling in adult flies. We found that *Aprt* transcripts were decreased by 29.5% on average (p<0.001) in *AdoR*[KGex] mutant heads (*Figure 6C*, left panel), while Aprt activity was even more decreased by 78.4% (p<0.001) compared to wild-type flies (*Figure 6C*, right panel). This effect likely results from the much increased level of adenosine uptake in the *AdoR* mutants (*Knight et al., 2010*), which would downregulate *Aprt* activity by a feedback mechanism. Overall, these data indicate that extracellular adenosine levels must be strongly decreased in Aprt mutant flies, both from the general reduction in adenosine levels and the increased expression of the Ent2 transporter. Although AdoR expression is not affected, AdoR signaling is therefore expected to be significantly reduced in *Aprt* mutants.

### *Aprt* mutants show epilepsy-like seizure behavior

A number of *Drosophila* mutants with disrupted metabolism or neural function show increased susceptibility to seizure and paralysis following strong mechanical or electrophysiological stimulation (*Fergestad et al., 2006*; *Parker et al., 2011*; *Kroll et al., 2015*). These mutants are called bang sensitive (BS) and commonly used as models of epileptic seizure (*Song and Tanouye, 2008*). Here we checked whether Aprt deficiency could trigger a BS phenotype. We observed that aged *Aprt* mutants (at 30 d a.E.) recovered slowly after a strong mechanical stimulation applied by vortexing the vial for 10 s at high speed. These flies took on average 17.3 s to recover and get back on their legs compared to 2.5 s for wild-type flies of the same age (p<0.01) (*Figure 7A*; see also *Figure 7—videos 1 and 2*). Some of the mutant flies appeared more deeply paralyzed as they did not spontaneously recover unless the vial was stirred, so their recovery time could not be scored. Hemizygous flies of the same age containing the *Aprt*[5] mutation over two partially overlapping genomic deficiencies covering *Aprt* also showed a BS behavior, with an average longer recovery time of 28.9 s and 33.2 s, respectively (p<0.001) (*Figure 7B*).

We then downregulated *Aprt* by RNAi in all cells with the *da-Gal4* driver to check if this could also induce BS behavior. As shown in *Figure 7C*, *da>Aprt*[RNAi] flies at 30 d a.E. indeed displayed seizure after mechanical shock, quite similar to that of the *Aprt*[5] mutants (22.7 s recovery time on average compared to 1.55 s and 0.72 s for the driver and RNAi controls, respectively, p<0.05). In contrast, inactivating *Aprt* selectively in neurons *(elav-Gal4)*, glial cells *(repo-Gal4)*, or muscles *(24B-Gal4)* did not induce a BS phenotype (*Figure 7—figure supplement 1*). This suggests that the BS phenotype requires *Aprt* knockdown in other cells or in several of these cell types. Finally, in contrast to the SING defect, we observed that adult-specific *Aprt* knockdown in all cells with *da-Gal4* for 3 d did not trigger

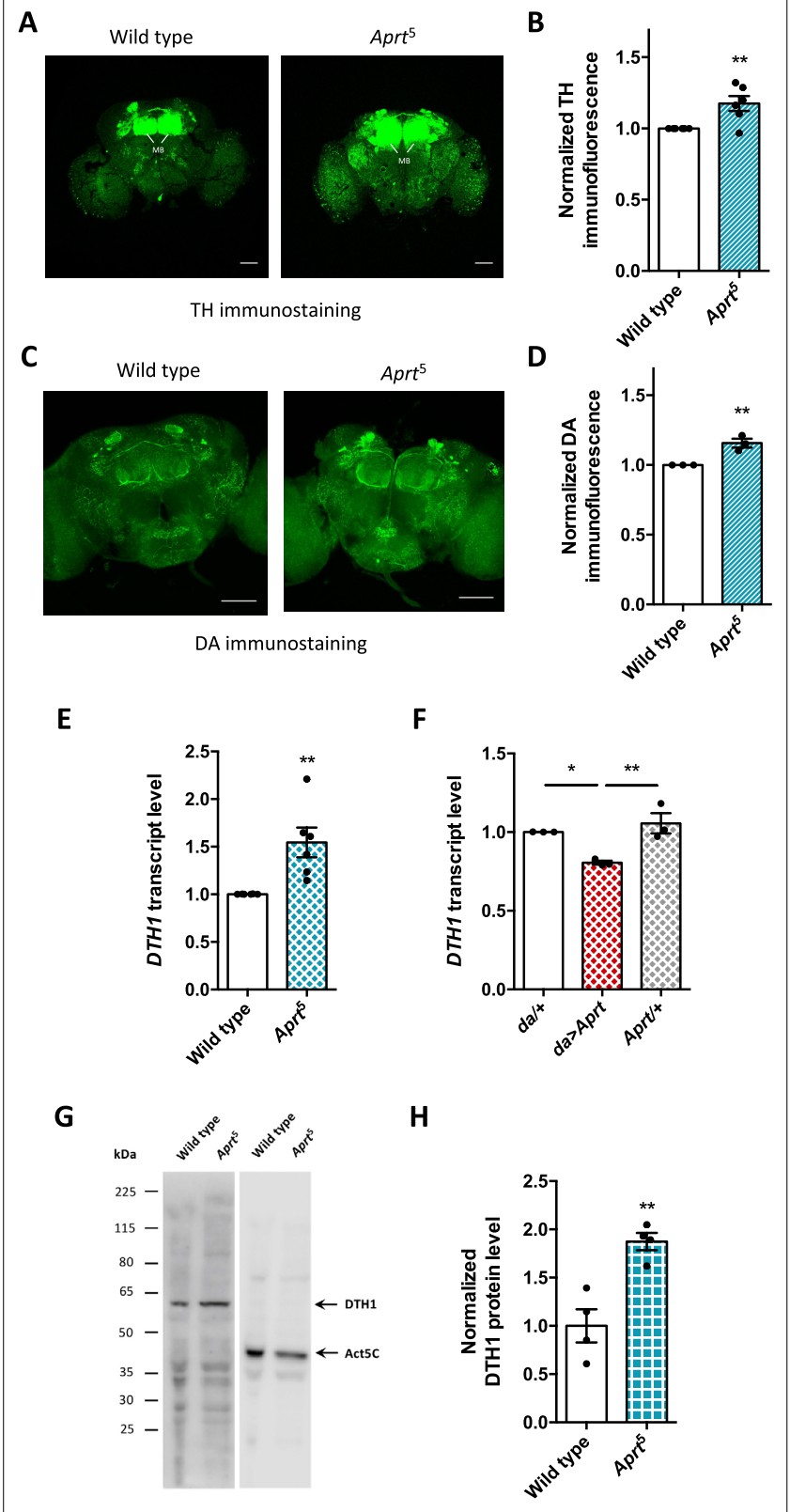

**Figure 5.** *Aprt* deficiency increases dopamine (DA) synthesis and content in the *Drosophila* brain.
(**A**) Representative confocal projections of tyrosine hydroxylase (TH)-immunostained whole-mount adult brains from wild-type flies and *Aprt*⁵ mutants. MB: mushroom body. Scale bars: 100 μm. (**B**) Quantification of TH immunofluorescence intensity normalized to the controls in the entire brain. 4–6 brains were dissected per

*Figure 5 continued on next page*

*Figure 5 continued*

experiment and genotype, and 6 independent experiments were performed (**p<0.01). (**C**) Representative confocal projections of DA immunostaining in whole-mount adult brains of wild-type and *Aprt⁵* mutants. Scale bars: 100 μm. (**D**) Quantification of DA immunofluorescence intensity over the entire brain showed a slight increase in DA content in *Aprt⁵* mutants compared to wild-type controls. Six brains were dissected per experiment and genotype, and three independent experiments were performed. Unpaired Student's *t*-test (**p<0.01). (**E**) mRNA levels of TH neuronal form *DTH1* are increased in *Aprt⁵* mutant heads compared to wild-type flies. Results of six independent RT-qPCR experiments carried out on 3–4 different RNA extractions from 20 to 30 male heads per genotype. Unpaired Student's *t*-test (**p<0.01). (**F**) Conversely, overexpressing Aprt in all cells with the *da-Gal4* driver (*da>Aprt*) reduced mRNA level of *DTH1* in heads compared to the driver (*da/+*) and effector (*Aprt/+*) controls. Mean of three independent experiments performed on three different RNA extractions from 20 to 30 male fly heads. One-way ANOVA with Tukey's *post hoc* test for multiple comparisons (*p<0.05, **p<0.01). (**G**) Representative western blot of wild-type and *Aprt⁵* mutant head extracts probed with anti-TH and anti-actin beta antibodies. (**H**) Quantification of DTH1 protein levels in adult wild-type and *Aprt⁵* mutant heads by western blots showed an increase in DTH1 protein level (60 kDa) in *Aprt⁵* mutants. Actin (Act5C, 42 kDa) was used as a loading control. Results are the mean of four determinations in two independent experiments. Unpaired Student's *t*-test (**p<0.01).

The online version of this article includes the following source data for figure 5:

**Source data 1.** Source data for *Figure 5A–F and H*.

**Source data 2.** Original files for the western blot analysis of *Figure 5G*.

**Source data 3.** Original files for the western blot analysis of *Figure 5G* with relevant bands and samples labeled.

a BS behavior (*Figure 7D*), indicating that the BS requires a longer downregulation of the gene or might be the consequence of a developmental defect.

## Administration of adenosine or *N⁶*-methyladenosine to *Aprt*-deficient flies prevents seizure

*Drosophila* disease models are advantageously tractable for drug screening in vivo (*Fernández-Hernández et al., 2016*; *Perrimon et al., 2016*). We thus administered several compounds related to purine metabolism to *Aprt⁵* flies to check if they can rescue neurobehavioral impairments (loco-motor defects and seizure). Feeding allopurinol at the same concentration used for uric acid rescue (100 μg/ml, *Figure 1B*), either in adults 5 d before the test or throughout all developmental stages, did not alter the BS phenotype (*Figure 7—figure supplement 2*), as was the case for the SING assay (*Figure 1D and E*). Similarly in humans, it has been shown that the daily intake of allopurinol, even from infancy, does not mitigate the neurobehavioral impairments in LND patients (*Marks et al., 1968*; *Torres et al., 2007a*; *Jinnah et al., 2010*; *Madeo et al., 2019*).

Then, we tried to supplement *Aprt* mutants with various purine compounds, including adenine, hypoxanthine, adenosine, and *N⁶*-methyladenosine (m⁶A), at 100 or 500 μM, either in adult stage 5 d before the test or throughout larval development plus 5 d before the test. None of these drugs was able to rescue the SING defect (*Figure 7—figure supplement 3*). In contrast and interestingly, administration of 500 μM adenosine or m⁶A during development rescued the BS phenotype of *Aprt⁵* mutants (*Figure 7E and F*). This further indicates that different mechanisms underpin SING alteration and BS behavior in *Aprt* mutants and provide another evidence that the BS may be caused by a developmental defect. The results also suggest that the lower adenosine levels of *Aprt* mutant flies could be at the origin of their BS.

## Neuronal expression of mutant HGPRT induces early locomotor defects and seizure behavior

In order to potentially develop another *Drosophila* model mimicking LND conditions, we generated new transgenic UAS lines to express in living flies either the human wild-type HGPRT (HGPRT-WT) or a pathogenic LND-associated mutant form of this protein (HGPRT-I42T), both isoforms being inserted at the same genomic docking site. These lines were validated by showing that they are transcribed at similar levels (*Figure 8A and B*). Enzymatic assays on adult extracts of flies expressing the wild-type form HGPRT-WT in all cells with *da-GAL4* showed significant HGPRT enzyme activity, while no activity

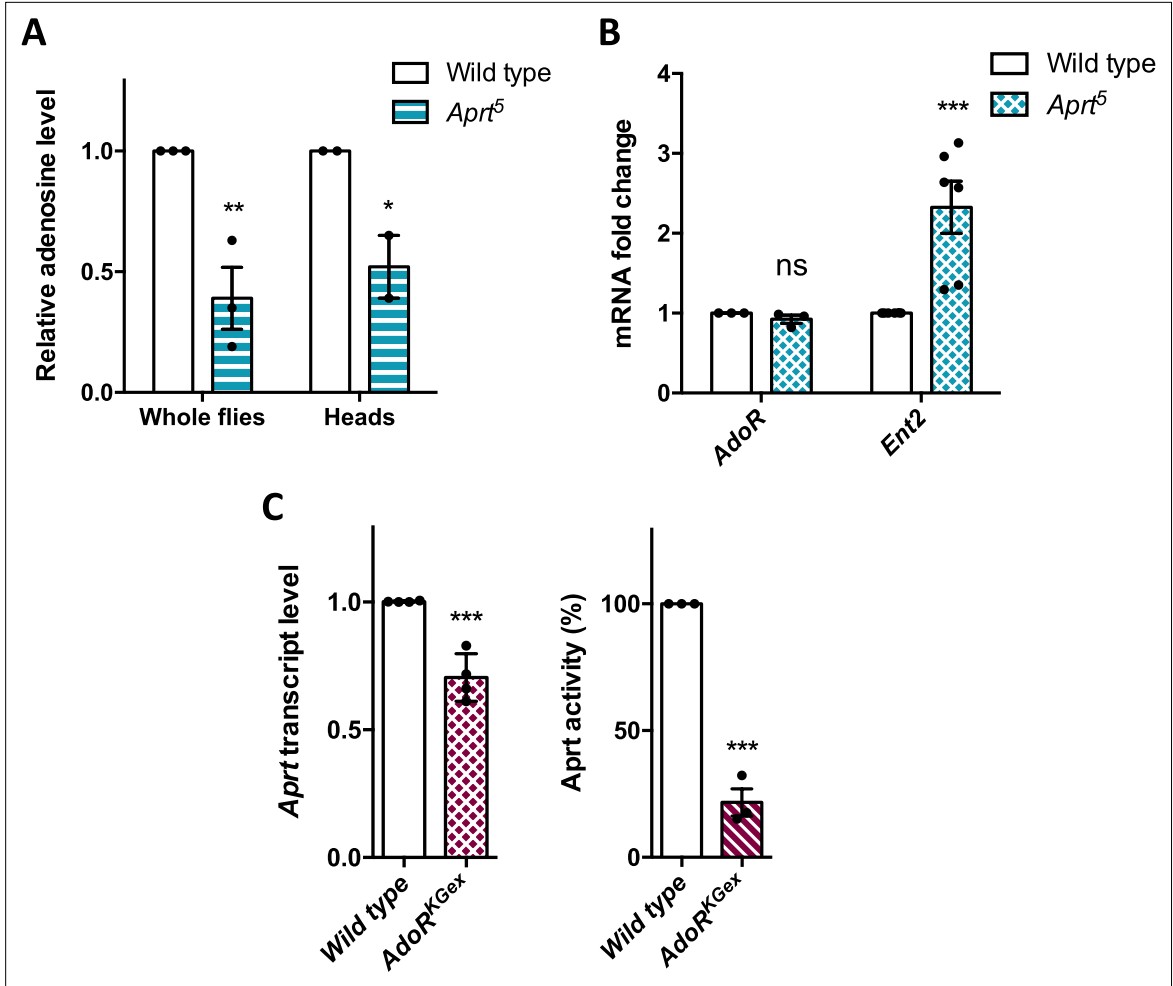

**Figure 6.** Relations between Aprt and molecular components of adenosinergic signaling. (**A, B**) Impacts of the lack of Aprt activity on the adenosinergic system. (**A**) Adenosine level was measured in whole flies or heads of *Aprt*[5] flies by ultra performance liquid chromatography (UPLC). Compared to wild-type flies, adenosine level was significantly reduced in the mutants. Results of three independent experiments performed with five males per genotype in triplicates and two independent experiments with 30 heads per genotype in triplicates. Two-way ANOVA with Sidak's *post hoc* test for multiple comparisons (*p<0.05; **p<0.01). (**B**) *Aprt*[5] mutation did not affect *AdoR* expression but induced a marked increase in adenosine transporter *Ent2* mRNA abundance. 3–6 different RNA extractions were performed on 20–30 male heads. Results of 3–6 independent experiments. Two-way ANOVA with Sidak's *post hoc* test for multiple comparisons (***p<0.001; ns: not significant). (**C**) Null *AdoR*[KGex] mutants showed decreased *Aprt* expression (left panel) and a stronger decrease in Aprt activity (right panel). Four independent RNA extractions were carried out on 20–30 male heads and four independent real-time PCR determinations were done per RNA sample. For Aprt activity, three independent determinations were performed on 20 whole flies per genotype. Unpaired Student's *t*-test (***p<0.001).

The online version of this article includes the following source data for figure 6:

**Source data 1.** Source data for *Figure 6A–C*.

was detected in driver and UAS control flies, and an 80.5% lower activity was detected in *Drosophila* expressing the mutant form HGPRT-I42T (*Table 2*).

We next analyzed the consequences of human HGPRT expression on the SING and BS behaviors. Interestingly, the pan-neuronal expression of mutant I42T isoform specifically induced a significant early locomotor defect at 15 d a.E. (PI = 0.71 vs 0.92 and 0.90 for the driver and effector controls, respectively, p<0.01) (*Figure 8C*) and a relatively small but robust BS behavior at 30 d a.E. (2.3 s average recovery time vs 0.64 s and 0.31 s for the driver and effector only controls, p<0.001) (*Figure 8D*). These defective phenotypes could not be seen when HGPRT-WT was expressed. Therefore, and remarkably, whereas wild-type HGPRT expression appears to be innocuous in *Drosophila*, we observed that the neuronal expression of a pathogenic LND-associated isoform triggered neurobehavioral impairments comparable to those of *Aprt*-deficient flies.

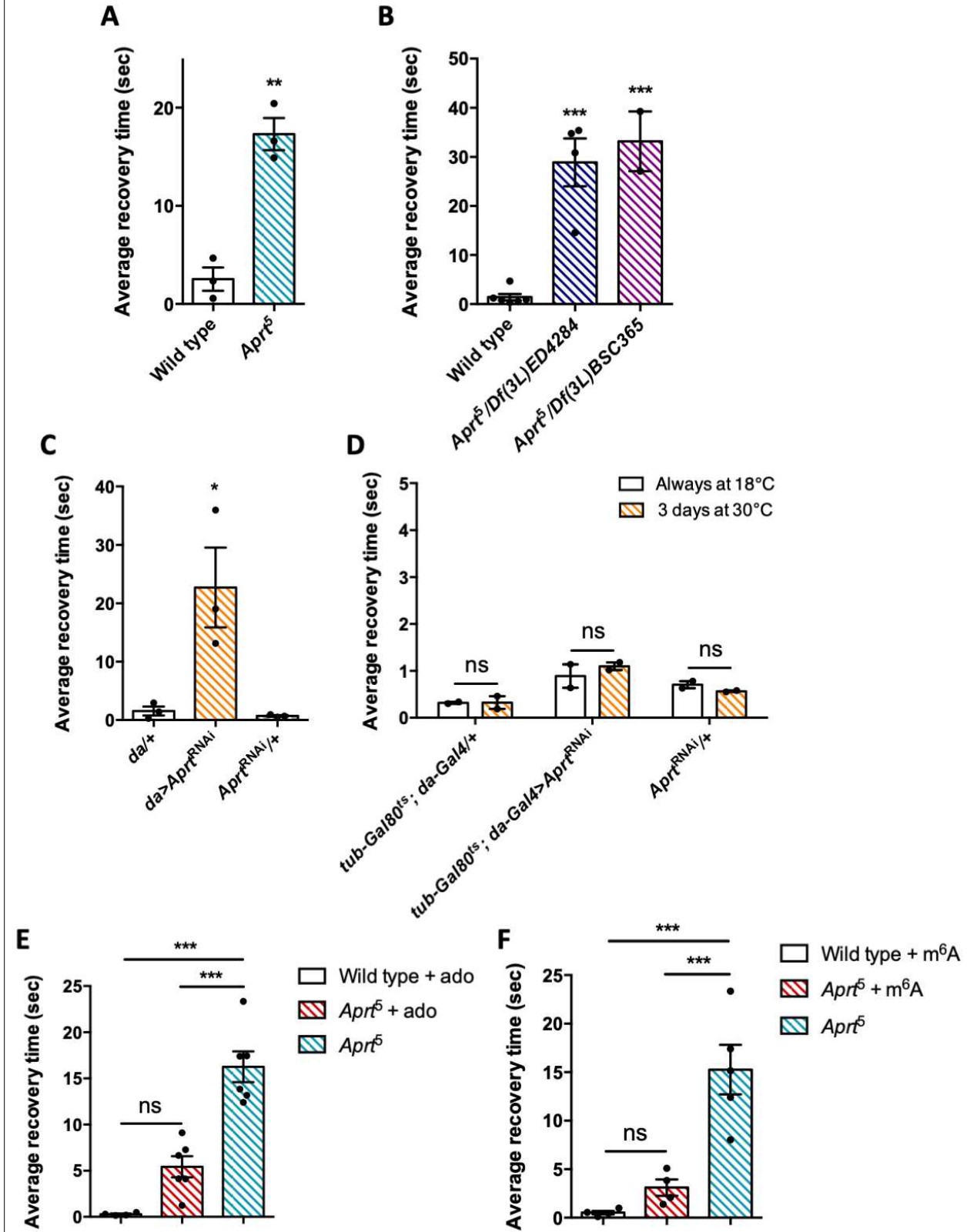

**Figure 7.** *Aprt* deficiency triggers a seizure-like phenotype. (**A**) At 30 days after eclosion (d a.E.), *Aprt⁵* mutants need a much longer time than wild-type flies to recover from a strong mechanical shock, showing a bang-sensitive (BS) paralysis comparable to tonic-clonic seizure. Results of three independent experiments performed on 50 flies per genotype. Unpaired Student's *t*-test; **p<0.01. (**B**) At 30 d a. E., hemizygous *Aprt⁵* mutants also showed a marked BS phenotype. Results of 2–4 independent experiments performed on 50 flies per genotype. One-way ANOVA with Dunnett's *post hoc* test for multiple

*Figure 7 continued on next page*

Figure 7 continued

comparisons (\*\*\*p<0.001). (**C**) RNAi-mediated downregulation of *Aprt* in all cells (*da>Aprt*[RNAi]) also led to BS phenotype in adults at 30 d a.E., but not with the driver and effector controls. Results of three independent experiments performed on 50 flies per genotype. One-way ANOVA with Tukey's *post hoc* test for multiple comparisons (\*p<0.05). (**D**) *Aprt* knockdown by RNAi during the adult stage for 3 d before the test was not sufficient to induce bang sensitivity, suggesting that this phenotype could be caused by a developmental defect or a longer downregulation of *Aprt*. Results of two independent experiments performed on 50 flies per genotype. Two-way ANOVA with Sidak's *post hoc* test for multiple comparisons; ns: not significant. (**E, F**) The BS phenotype of 30-day-old *Aprt*[5] mutants was rescued by feeding either 500 μM adenosine (ado) (**E**) or 500 μM $N^6$-methyladenosine (m[6]A) (**F**) during all developmental stages plus 5 d before the test. Results of four or six independent experiments performed on 50 flies per genotype. One-way ANOVA with Tukey's *post hoc* test for multiple comparisons (\*\*\*p<0.001, ns: not significant).

The online version of this article includes the following video, source data, and figure supplement(s) for figure 7:

**Source data 1.** Source data for *Figure 7A–F*.

**Figure supplement 1.** *Aprt* knockdown selectively in neurons, glia, or muscle cells did not induce bang sensitivity.

**Figure supplement 1—source data 1.** Source data for *Figure 7—figure supplement 1*.

**Figure supplement 2.** Administration of allopurinol does not rescue the bang sensitivity phenotype of *Aprt*-deficient mutants.

**Figure supplement 2—source data 1.** Source data for *Figure 7—figure supplement 2*.

**Figure supplement 3.** Administration of various purine compounds does not rescue the motricity defects of *Aprt*-deficient mutants.

**Figure supplement 3—source data 1.** Source data for *Figure 7—figure supplement 3*.

**Figure 7—video 1.** Bang sensitivity phenotype of Aprt-deficient flies: wild-type flies.

https://elifesciences.org/articles/88510/figures#fig7video1

**Figure 7—video 2.** Bang sensitivity phenotype of Aprt-deficient flies: *Aprt*[5] mutant flies.

https://elifesciences.org/articles/88510/figures#fig7video2

## Discussion

Over the past 35 years, several animal models of LND have been developed in rodents based on HGPRT mutation in order to better understand the cause of the disease and test potential therapeutic treatments (*Finger et al., 1988*; *Dunnett et al., 1989*; *Jinnah et al., 1993*; *Engle et al., 1996*; *Meek et al., 2016*; *Witteveen et al., 2022*). However, none of these models recapitulated the full LND syndrome and, particularly, the motor or neurobehavioral symptoms resulting from HGPRT deficiency. To date, the causes of the neurobehavioral impairments in LND are not yet clearly elucidated and the disease is still incurable (*Fu et al., 2014*; *Bell et al., 2016*; *López et al., 2020*; *Bell et al., 2021*; *Witteveen et al., 2022*). Here we used two different strategies to develop new models of this disease in *Drosophila*, a useful organism to conduct genetic and pharmacological studies. First, we show that *Aprt* deficiency induces both metabolic and neurobehavioral disturbances in *Drosophila*, similar to the loss of HGPRT, but not APRT, activity in humans. Secondly, we expressed an LND-associated mutant form of human HGPRT in *Drosophila* neurons, which also yielded behavioral symptoms. Our results suggest that the fruit fly can be used to study the consequences of defective purine recycling pathway and HGPRT mutation in the nervous system.

### *Aprt*-deficient flies replicate lifespan and metabolic defects caused by HGPRT deficiency

Flies that carry a homozygous null-mutation in *Aprt* develop normally and live until the adult stage (*Johnson and Friedman, 1983*). While a previous study reported that heterozygous *Aprt*/+ flies have an extended lifespan (*Stenesen et al., 2013*), we observed that homozygous *Aprt*[5] mutants have in contrast a significantly reduced longevity. The lack of HGPRT activity in LND also reduces lifespan expectancy, generally under 40 years of age for properly cared patients. *Stenesen et al., 2013* showed that, in *Drosophila*, dietary supplementation with adenine, the Aprt substrate, prevented the longevity extension conferred either by dietary restriction or heterozygous mutations of AMP biosynthetic enzymes. This suggests that lifespan depends on accurate adenine level regulation. It is possible that adenine could accumulate to toxic levels during aging in homozygous *Aprt* mutants, explaining their shorter lifespan. Alternatively, since AMP is the Aprt product, AMP-activated protein kinase (AMPK), an enzyme that protects cells from stresses inducing ATP depletion, could be less activated in Aprt mutants. Multiple publications explored the role of AMPK in lifespan regulation (*Sinnett and Brenman, 2016*) and downregulating AMPK by RNAi in adult fat body or muscles (*Stenesen*

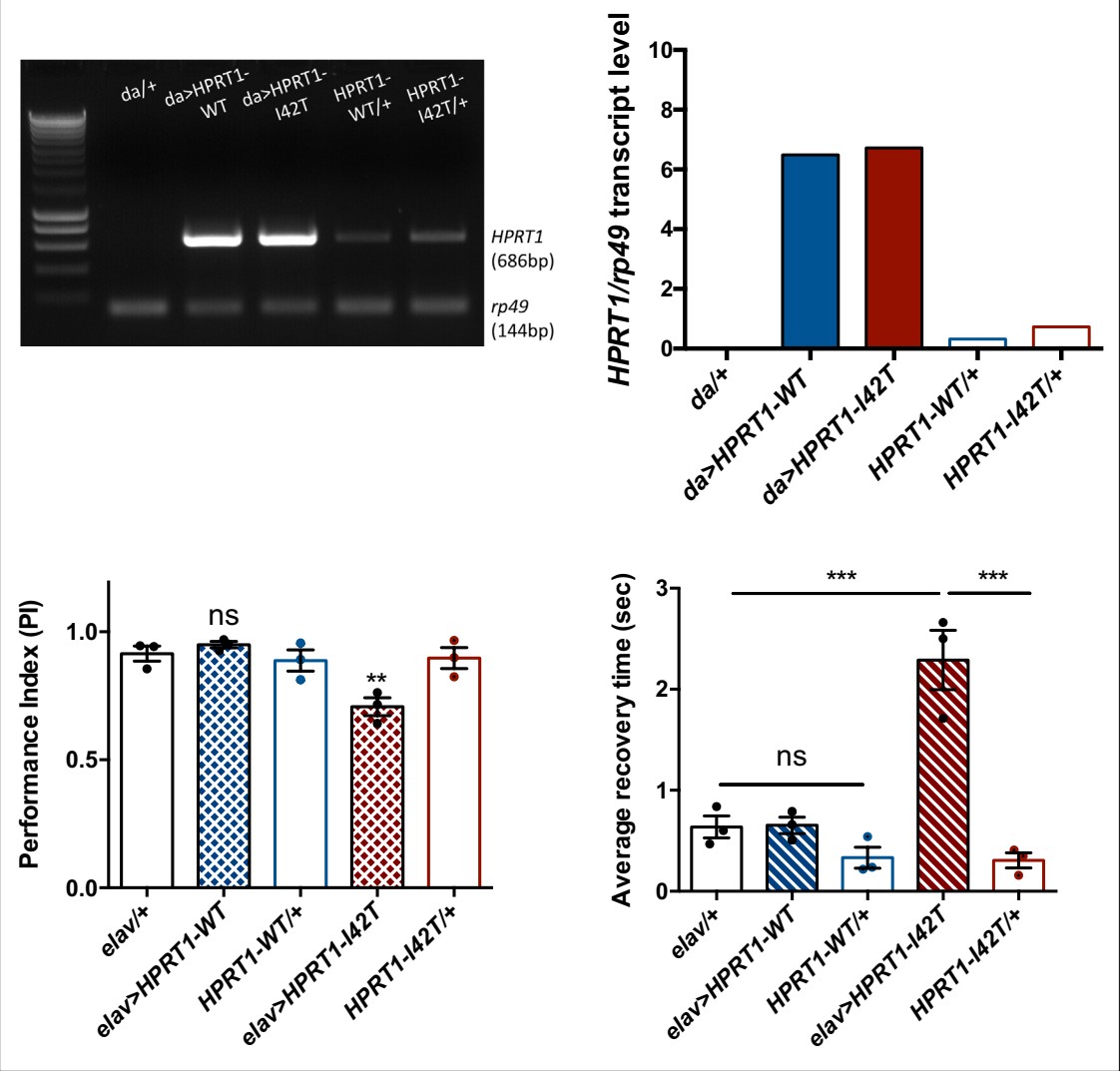

**Figure 8.** Expression of a pathogenic mutant isoform of human hypoxanthine-guanine phosphoribosyltransferase (HGPRT) induces neurobehavioral defects in flies. (**A, B**) Ubiquitous expression of human *HPRT1* with *da-Gal4*. (**A**) Amplification of human *HPRT1* transcripts detected by RT-PCR in head extracts of *da>HPRT1* WT and *da>HPRT1-I42T* flies. A band with lower intensity was also detected in the effector controls (*UAS-HPRT1-WT/+* and *UAS-HPRT1-I42T/+*), and not in the driver control (*da/+*), which indicates a small amount of driver-independent transgene expression. (**B**) Quantification of the previous experiment. (**C, D**) Expression of the Lesch–Nyhan disease (LND)-associated I42T isoform in all neurons (*elav>HPRT1-I42T*), but not of the wild-type form (*elav>HPRT1-WT*), induced an early SING defect at 15 d a.E. (**C**) and a BS phenotype at 30 d a.E. (**D**), compared to the driver (*elav/+*) and effector (*UAS-HPRT1-I42T/+*) only controls. Results of three independent experiments performed on 50 flies per genotype. One-way ANOVA with Tukey's *post hoc* test for multiple comparisons (**p<0.01; ***p<0.001; ns: not significant).

The online version of this article includes the following source data for figure 8:

**Source data 1.** Original file for the DNA gel electrophoresis of *Figure 8A*.

**Source data 2.** Original file for the DNA gel electrophoresis of *Figure 8A* with relevant bands and samples labeled.

**Source data 3.** Source data for *Figure 8B–D*.

et al., 2013), as well as its ubiquitous inactivation under starvation (*Johnson et al., 2010*), both reduced fly lifespan.

In humans, HGPRT deficiency induces hypoxanthine and guanine accumulation, resulting from lack of recycling, and increased de novo purine synthesis (*Harkness et al., 1988*; *Fu et al., 2015*; *Ceballos-Picot et al., 2015*). This in turn leads to uric acid overproduction that increases the risk for nephrolithiasis, renal failure, and gouty arthritis if not properly treated. In insects, the end product of purine catabolism is not uric acid but allantoin (*Figure 1—figure supplement 1*, step 20). However, urate oxidase, the enzyme that converts uric acid into allantoin, is specifically expressed in the Malpighi

**Table 2.** Hypoxanthine-guanine phosphoribosyltransferase (HGPRT) activity in transgenic flies expressing the wild-type or a Lesch–Nyhan disease (LND)-associated mutant form of human HPRT1.

| Genotypes | HGPRT activity (nmol/min/mg) |
|---|---|
| da/+ | 0 |
| da>HPRT1-WT | 13.88 ± 3.75 |
| da>HPRT1-I42T | 2.70 ± 1.44 |
| HPRT1-WT/+ | 0 |
| HPRT1-I42T/+ | 0 |

The online version of this article includes the following source data for table 2:

**Source data 1.** Source data for *Table 2*.

tubules, an excretory organ producing pre-urine (*Wallrath et al., 1990*). Uric acid could therefore accumulate in fly tissues and hemolymph if purine salvage pathway is impaired. Accordingly, we observed an increase in uric acid levels in *Drosophila Aprt* mutant heads, which could be reduced to normal levels by providing allopurinol, a xanthine oxidase inhibitor used to protect against renal failure in LND patients.

## Aprt is required in dopaminergic and mushroom body neurons for young fly motricity

We found that *Aprt*-null adult flies show reduced performance in the SING test, which monitors locomotor reactivity to startle and climbing ability. This phenotype appears at an early age, starting from 1 d a.E. The performance continued to decrease until 8 d a.E. and then appeared to stabilize up to 30 d a.E. This defect is quite different from the locomotor impairments described in *Drosophila* Parkinson's disease models, in which SING performance starts declining at around 25 d a.E. (*Feany and Bender, 2000*; *Riemensperger et al., 2013*; *Rahmani et al., 2022*). This phenotype of *Aprt*-deficient flies could be reminiscent of the early onset of motor symptoms in LND patients, which appear most often between 3 and 6 months of age (*Jinnah et al., 2006*). Interestingly, knocking down *Aprt* during 3 d only in adult flies also induced SING impairment, which argues against a developmental flaw. Although *Aprt* mutants walked slower than wild-type flies, they were no less active and their ATP levels were not different compared to controls, excluding a major failure in energy metabolism.

Downregulating *Aprt* in all neurons reproduced the locomotor defect of the *Aprt*[5] mutant, and *Aprt* mutant locomotion could be partially rescued by neuronal Aprt re-expression. The fact that rescue was not complete could be due to a dominant negative effect of the mutation, as suggested by enzymatic assays on extracts of heterozygous *Aprt*[5] mutants (*Figure 1—figure supplement 4*). Previous work from our and other laboratories showed that DA neurons control the SING behavior in *Drosophila* (*Feany and Bender, 2000*; *Friggi-Grelin et al., 2003*; *Riemensperger et al., 2011*; *Vaccaro et al., 2017*; *Sun et al., 2018*) and that PAM DA neurondegeneration induces SING defects in various Parkinson's disease models (*Riemensperger et al., 2013*; *Bou Dib et al., 2014*; *Tas et al., 2018*; *Pütz et al., 2021*). Here, we found that knocking down *Aprt* either in all DA neurons or only in the PAM DA neurons was sufficient to induce early SING defects. In contrast, knocking down Aprt with *TH-Gal4* that labels all DA neurons except for a major part of the PAM cluster did not induce SING defects, indicating that Aprt expression in subsets of PAM neurons is critical for this locomotor behavior.

Inactivating *Aprt* in all mushroom body neurons also induced a lower performance in the SING assay. This important brain structure receives connections from DA neurons, including the PAM, and is enriched in DA receptors (*Waddell, 2010*). We recently reported that activation of MB-afferent DA neurons decreased the SING response, an effect that requires signalization by the DA receptor dDA1/Dop1R1 in mushroom body neurons (*Sun et al., 2018*). We also observed a SING defect after knocking down *Aprt* either in all glial cells or more selectively in glial subpopulations expressing the glutamate transporter Eaat1, but not in the ensheathing glia. Astrocyte-like glial cells expressing *Eaat1* extend processes forming a thick mesh-like network around and inside the entire mushroom body neuropil (*Sinakevitch et al., 2010*). It could be hypothesized that SING also requires Aprt expression in these MB-associated astrocytes. However, re-expressing *Aprt* with *Eaat1-Gal4* did not lead to SING rescue in Aprt mutant background. This suggests that the presence of Aprt in neurons can somehow compensate for *Aprt* deficiency in glia, but the reverse is not true. In conclusion, proper startle-induced locomotion in young flies depends on Aprt activity in PAM DA and mushroom body neurons, and in *Eaat1*-expressing glial cells.

## Neuronal Aprt regulates spontaneous activity and sleep

Because sleep is regulated by the mushroom body as well as DA neurons in flies (*Artiushin and Sehgal, 2017*), we monitored spontaneous locomotor activity and sleep pattern of *Aprt*[5] mutants. Their activity profile appeared normal, with unaltered morning and evening anticipation, indicating that the circadian rhythms are surprisingly maintained in light-dark (LD) conditions in the absence of a functional purine recycling pathway (*Figure 4—figure supplement 1*). This experiment also highlighted that *Aprt*-deficient flies are hyperactive during the night, which suggests that they sleep less than wild-type flies. This could be confirmed by measuring their sleep pattern on video recordings. *Aprt* mutants show a reduced walking speed, sleep less during both day and night, and have difficulty in maintaining sleep. Downregulating *Aprt* selectively in neurons reproduced the sleep defect, whereas doing it in glial cells only had no effect. We have not attempted here to identify further the neuronal cells involved in this phenotype. Like for the SING behavior defect, it could involve PAM DA neurons as subpopulations of this cluster have been shown to regulate sleep in *Drosophila* (*Nall et al., 2016*). Therefore, the lack of purine recycling markedly disrupts sleep in *Aprt*-deficient flies, making them more active at night. This is strikingly comparable to young LND patients who have a much disturbed sleep time during the night (*Mizuno et al., 1979*).

## Lack of Aprt reduces adenosine signaling leading to DA neuron overactivation

We observed that *Aprt* deficiency did not decrease DA levels in the *Drosophila* brain. This prompted us to study another neuromodulator, adenosine, which is indirectly a product of Aprt enzymatic activity. The purine nucleoside adenosine is one of the building blocks of RNA and the precursor of ATP and cAMP, but is also the endogenous ligand of adenosine receptors that modulate a wide range of physiological functions. In brain, adenosine regulates motor and cognitive processes, such as the sleep-wake cycle, anxiety, depression, epilepsy, memory, and drug addiction (*Soliman et al., 2018*). In normal metabolic conditions, adenosine is present at low concentrations in the extracellular space and its level is highly regulated, either taken up by cells and incorporated into ATP stores or deaminated by adenosine deaminase into inosine. In mammals, several nucleoside transporters mediate the uptake of adenosine and other nucleosides into cells, named equilibrative and concentrative nucleoside transporters, respectively (*Gray et al., 2004*; *Boswell-Casteel and Hays, 2017*; *Pastor-Anglada and Pérez-Torras, 2018*).

We observed a marked reduction in adenosine levels in *Aprt* mutant flies. While we did not observe any alteration in the transcript levels of *AdoR*, the gene coding for the only adenosine receptor in *Drosophila*, transcript levels of an adenosine transporter, Ent2, were increased more than twofold in *Aprt* mutants. Interestingly, one paper reported a similar strong increase in *Ent2* expression in *AdoR* mutant flies (*Knight et al., 2010*). These results suggest that AdoR signaling is less activated in *Aprt*-deficient flies compared to controls. We also observed that the lack of AdoR decreased Aprt expression and activity in *Drosophila*, possibly from increased *Ent2* expression and so higher adenosine influx which could downregulate Aprt by a feedback mechanism.

Adenosine and DA receptors are known to interact closely in mammals (*Franco et al., 2000*; *Kim and Palmiter, 2008*). A previous study performed in *Drosophila* larvae showed that an increase in astrocytic $Ca^{2+}$ signaling can silence DA neurons through AdoR stimulation by a mechanism potentially involving the breakdown of released astrocytic ATP into adenosine (*Nall et al., 2016*). We previously showed that DA neuron activation can decrease fly performance in the SING test (*Sun et al., 2018*). Fly nocturnal hyperactivity can also be caused by an increase in DA signaling (*Kumar et al., 2012*; *Lee et al., 2013*), in accordance with our observation that Aprt-deficient flies sleep less and are more active during the night. Therefore, both the locomotor and sleep defects induced by the lack of Aprt activity could be explained by DA neuron overactivation that would result from reduced adenosinergic signaling.

Adenosine has been proposed before to be involved in neurological consequences of LND (*Visser et al., 2000*). Adenosine transport is decreased in peripheral blood lymphocytes of LND patients (*Torres et al., 2004*), as well as $A_{2A}$ adenosine receptor mRNA and protein levels (*García et al., 2009*; *García et al., 2012*). In a murine LND model, adenosine $A_1$ receptor expression was found to be strongly increased and that of $A_{2A}$ slightly decreased, while $A_{2B}$ expression was not affected (*Bertelli et al., 2006*). Chronic administration of high doses of caffeine, an adenosine receptor antagonist,

caused self-injurious behavior in rats (*Miñana et al., 1984*; *Jinnah et al., 1990*). Moreover, central injection of an adenosine agonist is sufficient to prevent self-mutilation induced by dopaminergic agonist administration in neonatally 6-OHDA lesioned rats (*Criswell et al., 1988*).

## Adenosine or N6-methyladenosine supplementation rescues the epilepsy behavior of *Aprt* mutants

LND is characterized by severe behavioral troubles, including dystonia, spasticity, and involuntary movements (choreoathetosis). Some patients can also show an epileptic disorder (*Jinnah et al., 2006*; *Madeo et al., 2019*). In flies, the BS test is often used to model epileptic seizures (*Song and Tanouye, 2008*; *Parker et al., 2011*). Here, we observed that 30-day-old *Aprt* mutant flies show a transient seizure-like phenotype after a strong mechanical shock. Although seizure duration appeared shorter in *Aprt*[5] than in typical BS mutants such as *bss*, at least one other BS mutant, *porin*, was reported to have similar short recovery times as *Aprt*-deficient flies (*Graham et al., 2010*). Previous works demonstrated that BS is linked to neuronal dysfunction in *Drosophila* (*Parker et al., 2011*; *Kroll and Tanouye, 2013*; *Kroll et al., 2015*; *Saras and Tanouye, 2016*). Knocking down *Aprt* in specific cells such as neurons, glia, or muscles did not trigger this phenotype, suggesting that Aprt must be inactivated in several cell types to induce seizure.

Interestingly, knocking down *Aprt* by RNAi in all cells during development also induced the BS behavior, but not for 3 d only in adult flies, at variance with the SING phenotype. We have fed the mutants with a diet supplemented with various compounds involved in purine metabolism, including allopurinol, adenine, hypoxanthine, adenosine, or its analog N6-methyladenosine (m⁶A) either throughout larval development or in adult stage. Only adenosine and m⁶A, ingested during development, rescued the BS behavior. This suggests that loss of Aprt induces a lack of adenosine in the developing nervous system, as we observed in adult flies (*Figure 6*), which may alter neural circuits controlling BS behavior in adults. The adenosine analog m⁶A cannot be incorporated into nucleic acids and is excreted in the urine (*Schram, 1998*; *Batista, 2017*). The rescuing effect we observed with m⁶A suggests thereby that both this compound and adenosine are required as adenosine receptor agonists or allosteric regulators during development, rather than nucleotide precursors.

Adenosine can strongly inhibit cerebral activity and its role as endogenous anticonvulsant and seizure terminator is well established in humans (*Boison, 2005*; *Masino et al., 2014*; *Weltha et al., 2019*). Conversely, deficiencies in the adenosine-based neuromodulatory system can contribute to epileptogenesis. For instance, increased expression of astroglial adenosine kinase (ADK), which converts adenosine into AMP, leads to a reduction in brain adenosine level that plays a major role in epileptogenesis (*Weltha et al., 2019*). Hence, therapeutic adenosine increase is a rational approach for seizure control. Our observation that feeding adenosine or its derivative m⁶A rescued the seizure-like phenotype of *Aprt* mutant flies further suggests that adenosinergic signaling has partly similar functions in the fly and mammalian brains. In addition, the decrease in adenosine levels we observed in *Aprt* mutants could result from enhanced ADK activity that would compensate for the lack of Aprt-produced AMP.

We and others recently observed that m⁶A and related compounds sharing an adenosine moiety are able to rescue the viability of LND fibroblasts and neural stem cells derived from induced pluripotent stem cells (iPSCs) of LND patients cultured in the presence of azaserine, a potent blocker of de novo purine synthesis (Petitgas and Ceballos-Picot, unpublished results; *Ruillier et al., 2020*). Like in flies again, allopurinol was not capable of rescuing the cell viability in this in vitro model. The similarity of these results increases confidence that *Aprt*-deficient *Drosophila* could be used as an animal model of LND.

## Expression of mutant HGPRT triggers locomotor and seizure phenotypes

How HGPRT deficiency can cause such dramatic neurobehavioral troubles in LND patients still remains a crucial question. To date, cellular (*Smith and Friedmann, 2000*; *Torres et al., 2004*; *Ceballos-Picot et al., 2009*; *Cristini et al., 2010*; *Guibinga et al., 2012*; *Sutcliffe et al., 2021*) and rodent (*Finger et al., 1988*; *Dunnett et al., 1989*; *Jinnah et al., 1993*; *Meek et al., 2016*; *Witteveen et al., 2022*) models only focused on the consequences of HGPRT deficiency to phenocopy the disease. Such an approach was justified by the fact that the lower the residual activity of mutant HGPRT, the more

severe the symptoms are in patients (*Fu and Jinnah, 2012*, *Fu et al., 2014*). However, it could be conceivable that part of these symptoms result from compensatory physiological mechanisms or a deleterious gain-of-function conferred to the HGPRT protein by the pathogenic mutations. Here, we observed that neuronal expression of mutant human HGPRT-I42T, which expresses a low enzymatic activity, but not the wild-type HGPRT protein, induced early locomotor defects in young flies and BS in older flies, similarly to the defects induced by Aprt deficiency. This suggests a potential neurotoxicity of the pathological mutant form of HGPRT, which could be related to disturbances in purine metabolism or other signaling pathways. The human mutant form might not be properly degraded and accumulate as aggregates, potentially exerting neuronal toxicity. Such an approach opens interesting perspectives to better understand the endogenous function of HGPRT and its pathogenic forms. Indeed, the identification of a potential inherent neurotoxicity of defective forms of human HGPRT is a new element, which could be explored in further work in the fly and also in rodent models.

## A new model of LND in an invertebrate organism?

LND, a rare X-linked metabolic disorder due to mutations of the *HPRT1* gene, has dramatic neurological consequences for affected children. To date, no treatment is available to abrogate these troubles, and no fully satisfactory in vivo models exist to progress in the understanding and cure of this disease. HGPRT knockout rodents do not show comparable motor and behavioral troubles, which makes these models problematic for testing new therapeutic treatments. *Drosophila* does not express HGPRT-like activity and our phylogenetic analysis established that no HGPRT homolog is present in *D. melanogaster* (*Figure 1—figure supplement 2* and *Figure 1—figure supplement 3*), confirming that Aprt is the only enzyme of the purine salvage pathway in this organism. APRT and HGPRT are known to be functionally and structurally related. Both human APRT and HGPRT belong to the type I PRTases family identified by a conserved phosphoribosyl pyrophosphate (PRPP) binding motif, which is used as a substrate to transfer phosphoribosyl to purines. This binding motif is only found in PRTases from the nucleotide synthesis and salvage pathways (*Sinha and Smith, 2001*). Moreover, the purine substrates adenine, hypoxanthine, and guanine share the same chemical skeleton and APRT can bind hypoxanthine, indicating that APRT and HGPRT also share similarities in their substrate binding sites (*Ozeir et al., 2019*).

Here, we find that *Aprt* mutant flies show symptoms partly comparable to the lack of HGPRT in humans, including increase in uric acid levels, reduced longevity, and various neurobehavioral defects such as early locomotor decline, sleep disorders, and epilepsy behavior. This animal model therefore recapitulates both salvage pathway disruption and motor symptoms, as observed in LND patients. Moreover, our results highlight that *Aprt* deficiency in *Drosophila* has more similarities with HGPRT than APRT deficiency in humans. *Aprt* mutant flies also show a disruption of adenosine signaling, and we found that adenosine itself or a derivative compound can relieve their epileptic symptoms. Finally, neuronal expression of a mutant form of human HGPRT that causes LND also triggers abnormalities in fly locomotion and seizure-like behavior, which has not been documented to date in other models. The use of *Drosophila* to study LND could therefore prove valuable to better understand the link between purine recycling deficiency and brain functioning and carry out drug screening in a living organism, paving the way toward new improvements in curing this dramatic disease.

# Materials and methods
## *Drosophila* culture and strains

Flies were raised at 25°C on standard cornmeal-yeast-agar medium supplemented with methyl hydroxybenzoate as an antifungal under a 12 hr:12 hr LD cycle. The *Drosophila* mutant lines were obtained either from the Bloomington Drosophila Stock Center (BDSC), the Vienna Drosophila Resource Center (VDRC) or our own collection: *Aprt⁵* (*Johnson and Friedman, 1983*) (BDSC #6882), *Df(3L)ED4284* (BDSC #8056), *Df(3L)BSC365* (BDSC #24389), *UAS-Aprt*^RNAi (VDRC #106836), *AdoR*^KG03964ex (*Wu et al., 2009*) here named *AdoR*^KGex (BDSC #30868), and the following Gal4 drivers: *238Y-Gal4*, *24B-Gal4*, *da-Gal4*, *Eaat1-Gal4* (*Rival et al., 2004*), *elav-Gal4* (*elav*^C155), *MZ0709-Gal4*, *NP6510-Gal4*, *NP6520-Gal4*, *R58E02-Gal4* (*Liu et al., 2012*), *R76F05-Gal4*, *repo-Gal4*, *TH-Gal4* (*Friggi-Grelin et al., 2003*), *TRH-Gal4* (*Cassar et al., 2015*), *tub-Gal80*^ts , and *VT30559-Gal4*. The Canton-S line was used as wild-type control. The simplified driver>effector convention was used to indicate genotypes in figures, for

example, *elav>Aprt*[RNAi] for *elav-Gal4; UAS-Aprt*[RNAi]. In some experiments, to restrict RNAi-mediated *Aprt* inactivation at the adult stage, we have used the TARGET system (*McGuire et al., 2003*). Flies were raised at 18°C (permissive temperature) where Gal4 transcriptional activity was inhibited by *tub-Gal80*[ts], and shifted to 30°C (restrictive temperature) for 3 d before the test to enable Gal4-driven *Aprt*[RNAi] expression.

## Construction of transgenic lines

To generate a *UAS-Aprt* strain, a 549 bp *Aprt* insert containing the coding sequence was PCR amplified from the ORF clone BS15201 (Drosophila Genomics Resource Center, Bloomington, IN) using primers with added restriction sites (in bold type): forward 5′-AGGGAATTGG**GAATTC**GTTATCAGTCGA CATGAGCCC, reverse 5′-ACAAAGATCC**TCTAGA**TCTAGAAAGCTTTCAGTACTTAATG. After digestion with EcoRI and XbaI, the *Aprt* cDNA was subcloned into the *pUASTattB* vector (*Bischof et al., 2007*) using the In-Fusion HD Cloning Kit (Takara Bio, Kyoto, Japan) according to the manufacturer's instructions, and the insertion verified by sequencing (Eurofins Genomics, Ebersberg, Germany). The construction was sent to BestGene (Chino Hills, CA, USA) for *Drosophila* germline transformation into the attP14 docking site on the 2d chromosome. The *UAS-Aprt* line yielding the strongest expression was selected and used in the experiments.

A clone containing the human wild-type *HPRT1* cDNA was kindly provided to us by Prof. Hyder A. Jinnah (Emory University, GA). We constructed the *HPRT1-I42T* cDNA from this clone by site-directed mutagenesis using the QuikChange II Site-Directed Mutagenesis Kit (Agilent Technologies, Santa Clara, CA). Primers sequences used to create the mutation were: forward 5'-CAGTCCTGTCCA TA**G**TTAGTCCATGAGGAATAAACACCCT and reverse 5'-AGGGTGTTTATTCCTCATGGACTAA**C**TATG GACAGGACTG (the bases modified to create the mutation are in bold type). The cDNA obtained was verified by sequencing. Then, the 657 bp *HPRT1-WT* and *HPRT1-I42T* inserts were PCR amplified using primers with added restriction sites (in bold type) and *Drosophila* translation start consensus sequence: forward 5′-AGGGAATTGG**GAATTC**AAGAAGGAGAT<u>ACAAA</u>ATGGC and reverse 5′-ACAA AGATCC**TCTAGA**GCTCGGATCCTTATCATTAC (the bases modified to match the *Drosophila* translation initiation consensus sequence are underlined). They were subcloned into *pUASTattB* and verified by sequencing. The transgenes were sent to BestGene for *Drosophila* transformation and inserted into the attP40 docking site on the 2d chromosome.

## Sequencing of *Aprt*[5]

For sequencing of the *Aprt*[5] cDNA, total RNA was isolated by standard procedure from 20 to 30 heads of homozygous *Aprt*[5] flies and reverse transcribed using oligo d(T) primers (PrimeScript RT Reagent Kit, Takara Bio). At least 750 ng of the first-strand cDNA was amplified by PCR in 20 µl of reaction mixture using PrimeStar Max DNA polymerase (Takara Bio) with a Techne Prime Thermal Cycler apparatus (Bibby Scientific, Burlington, NJ). The program cycles included 10 s denaturation at 98°C, 10 s annealing at 55°C, and 10 s elongation at 72°C, repeated 35 times. 1 µl of the PCR product was amplified again with the same program, in 30 µl of reaction mixture. After elution on 1% agarose gel, DNA were purified using the Wizard SV Gel and PCR Clean-Up System protocol (Promega, Madison, WI) according to the manufacturer's instructions. Finally, 7.5 µl of purified DNA were sent with 2.5 µl of primers (forward and reverse in separate tubes) for sequencing (Eurofins Genomics).

## Phylogenetic analyses

HGPRT homologs were identified by BlastP searching the NCBI GenBank non-redundant protein database (last October 2019 version). A subset of interest was selected for phylogenetic analyses. For *Figure 1—figure supplement 2*, multiple sequence alignment was performed using MAFFT (-ensi) (*Katoh and Standley, 2013*). The confidence of aligned residues was assessed using the TCSindex (*Chang et al., 2014*) only columns with TCS index 7–9 (on a 0–9 scale) were retained. ProtTest v3.2 (*Darriba et al., 2011*) was used to assess the best model fitting of the data. Maximum likelihood tree was inferred in IQ-TREE (*Trifinopoulos et al., 2016*), under the LG + R6 model. Bayesian inference was carried out in PhyloBayes v. 3.3 (*Lartillot et al., 2009*), under the LG + Γ4 model. For *Figure 1— figure supplement 3*, the whole analysis was performed in SeaView 5.0.4 (*Gouy et al., 2010*). Multiple sequence alignment was performed using MUSCLE (*Edgar, 2004*). Maximum likelihood tree

was inferred using PhyML 3.0 (*Guindon et al., 2010*) (under the LG + $\Gamma$4 model; best of NNI and SPR tree searching option; invariable sites optimized).

## Lifespan assay

Longevity study was performed as previously described (*Riemensperger et al., 2011*). *Drosophila* adult males were collected within 24 hr of emergence and maintained on standard medium at 25°C under a 12:12 hr LD cycle. They were transferred into fresh bottles every 2–3 d, and the number or surviving flies was scored. Also, 50 flies per bottle in triplicate were tested for each genotype and the experiment was done three times.

## Startle-induced negative geotaxis (SING)

SING assays were performed as previously described (*Rival et al., 2004*; *Riemensperger et al., 2011*; *Sun et al., 2018*). For each genotype, 50 adult males divided into five groups of 10 flies were placed in a vertical column (25 cm long, 1.5 cm diameter) with a conic bottom end and left for about 30 min for habituation. Then, columns were tested individually by gently tapping them down (startle), to which flies normally responded by climbing up. After 1 min, flies having reached at least once the top of the column (above 22 cm) and flies that never left the bottom (below 4 cm) were counted. Each fly group was assayed three times at 15 min intervals. The PI for each column is defined as $\frac{1}{2}[1 + (n_{top}-n_{bot})/n_{tot}]$, where $n_{tot}$ is the total number of flies in the column, $n_{top}$ the number of flies at the top, and $n_{bot}$ the number of flies at the bottom after 1 min. SING was performed at 1, 8, 10, and 30 d a.E.

## Spontaneous locomotion and sleep monitoring

Spontaneous locomotor activity was recorded as previously described (*Vaccaro et al., 2017*) using *Drosophila* activity infrared beam monitors (DAM, TriKinetics Inc, Waltham, MA) placed in incubators at 25°C equipped with standard white light. Eight-day-old male flies were maintained individually for 5–6 d under 12:12 hr LD cycle in 5 × 65 mm glass tubes containing 5% sucrose, 1.5% agar medium. Data analysis was performed with the FaasX software (*Klarsfeld et al., 2003*). Histograms represent the distribution of the activity through 24 hr in 30 min bins, averaged for 32 flies per genotype over 4–5 cycles.

For sleep monitoring, 2–4-day-old virgin female flies were transferred individually into 5 × 65 mm glass tubes containing standard food and their movements were recorded for up to 5 d using DAM infrared beam monitors (TriKinetics Inc) or the *Drosophila* ARousal Tracking (DART) video system (*Faville et al., 2015*), in a 12 hr:12 hr LD cycle with 50–60% humidity. Control and experimental flies were recorded simultaneously. Each experiment included at least 14 flies for each condition and was repeated 2–3 times with independent groups of flies. Fly sleep, defined as periods of immobility lasting more than 5 min (*Huber et al., 2004*), was computed with a Microsoft Excel macro for the infrared beam data and with the DART software for the video data. Distribution and homogeneity as well as statistical group comparisons were tested using Microsoft Excel plugin software StatEL. The p-value shown is the highest obtained among *post hoc* comparisons and means ± SEM were plotted.

## Immunohistochemistry

Brains of 8–10-day-old adult flies were dissected in ice-cold $Ca^{2+}$-free *Drosophila* Ringer's solution and processed for whole-mount anti-TH or anti-DA immunostaining as previously described (*Riemensperger et al., 2011*; *Cichewicz et al., 2017*). The primary antibodies used were mouse monoclonal anti-TH (1:1000, Cat# 22941, ImmunoStar, Hudson, WI) and mouse monoclonal anti-DA (1:100, Cat# AM001, GemacBio, Saint-Jean-d'Illac, France). The secondary antibody was goat anti-mouse conjugated to Alexa Fluor 488 (1:1000, Thermo Fisher Scientific, Waltham, MA). Brains were mounted with antifade reagent, either Prolong Gold (Thermo Fisher Scientific) or, alternatively, 65% 2,2'-thiodiethanol (Sigma-Aldrich, St. Louis, MI) (*Cichewicz et al., 2017*) for DA staining. Images were acquired on a Nikon A1R confocal microscope with identical laser, filter, and gain settings for all samples in each experiment. Immunofluorescence intensity levels were quantified using the Fiji software. Experiments were repeated independently at least three times on 4–6 brains per genotype.

## Bang sensitivity test

Bang sensitivity assays were performed as previously described (*Howlett et al., 2013*). 30-day-old males were divided into five groups of 10 flies under $CO_2$ and allowed to recover overnight. The following day, each group was placed in a vial without food and after 20 min of habituation, the vials were stimulated vigorously with a vortex mixer for 10 s at 2500 rpm. The recovery time was measured for each fly, from the end of the stimulation until they reached a normal standing position. Results are the mean of the recovery time for at least 50 flies per genotype.

## Drug administration

Allopurinol (A8003, Sigma-Aldrich) was diluted in standard medium at 100 µg/ml and flies were placed for 5 d on this medium before metabolite extraction. Adenine, adenosine, and hypoxanthine (A2786, A9251, and H9377, Sigma-Aldrich) or $N^6$-methyladenosine ($m^6A$) (QB-1055, Combi-Blocks, San Diego, CA) were diluted in fly food medium at 500 µM. Parents were allowed to lay eggs on this medium in order to have exposition to the drug throughout all larval development of the progeny. Adults were collected and placed in normal medium until 5 d before the test, when they were placed again in food supplemented with adenosine or $m^6A$ at the same concentrations.

## Uric acid quantification

For purine metabolite extraction, 40 heads from 8-day-old flies were ground in 80% ethanol, heated for 3 min at 80°C, and evaporated in a Speedvac apparatus. Dried residues were resuspended in MilliQ water and total protein content of each homogenate was measured with the Pierce BCA Protein Assay Kit (Thermo Fisher Scientific). 20 µl of each sample was injected into a 25 cm × 4.6 mm C18 Nucleosil column with 5 µm particles (Interchim, Montluçon, France). Purine metabolites were detected by an Agilent 1290 Infinity HPLC system coupled with a diode array detector as recommended by the ERDNIM advisory document. Seven wavelengths were used for detection (230, 240, 250, 260, 266, 270, and 280 nm) in order to have the spectrum of each compound for identification in addition to the retention time. The mobile phases contained 0.05 M monopotassium phosphate pH 5 and 65% (v/v) acetonitrile. The flux was fixed at 1 ml/min. For analysis, the maximum height of the compound was normalized to protein content and compared to the control genotype.

## ATP assay

ATP level was measured by bioluminescence using the ATP Determination Kit (A22066, Thermo Fisher Scientific) on a TriStar 2 Spectrum LB942S microplate reader (Berthold Technologies, Bad Wildbad, Germany) according to the manufacturer's instructions. Samples were prepared as described previously (*Fergestad et al., 2006*). Briefly, 30 heads or 5 thoraces from 8-day-old flies were homogenized in 200 µl of 6 M guanidine-HCl to inhibit ATPases, boiled directly 5 min at 95°C, and then centrifuged for 5 min at 13,000 × *g* and 4°C. Total protein content of each supernatant was measured with the BCA Protein Assay Kit. Each supernatant was then diluted at 1:500 in TE buffer pH 8 and 10 µl were placed in a 96-well white-bottom plate. The luminescent reaction solution (containing D-luciferine, recombinant firefly luciferase, dithiothreitol, and reaction buffer) was added to each well with an injector and high-gain 1 s exposure glow reads were obtained at 15 min after reaction initiation. Results were compared to a standard curve generated with known ATP concentrations, and final values were normalized to the protein content.

## Enzyme activity assay

HGPRT and APRT activities were assessed in *Drosophila* by adapting the methods previously established for human cells (*Cartier and Hamet, 1968*; *Ceballos-Picot et al., 2009*). Twenty whole male flies were homogenized in 250 µl of 110 mM Tris, 10 mM $MgCl_2$ pH 7.4 (Tris-$MgCl_2$ buffer), immediately frozen and kept at least one night at –80°C before the assay. After 5 min of centrifugation at 13,000 × *g*, total protein content of each supernatant was measured with the BCA Protein Assay Kit to normalize activity level. Kinetics of [$^{14}$C]-hypoxanthine conversion to IMP (or in some cases [$^{14}$C]-guanine conversion to GMP), and [$^{14}$C]-adenine conversion to AMP were assessed for HGPRT and Aprt assays, respectively. Compositions of reaction mediums were 25 µl of a radioactive solution made with 38 µl of [$^{14}$C]-hypoxanthine (25 µCi/ml) diluted in 1 ml of 1.2 mM cold hypoxanthine (or in some cases 38 µl of [$^{14}$C]-guanine [25 µCi/ml] diluted in 1 ml of 1.2 mM cold guanine), for HGPRT

assay, or 25 µl of a radioactive solution made with 75 µl of [$^{14}$C]-adenine (50 µCi/ml) diluted in 1 ml of 1.2 mM cold adenine for Aprt assay, and a volume of fly extract equivalent to 200 µg protein diluted in Tris-MgCl$_2$ buffer to 150 µl. Reactions were monitored at 37°C and started by adding 25 µl of the co-factor 5-phosphoribosyl-1-pyrophosphate (PRPP) at 10 mM. After 6, 12, 24, and 36 min, 40 µl of each pool was placed in a tube containing either 25 µl of HIE (3 mM hypoxanthine, 6 mM IMP, 200 mM EDTA) or AAE (3 mM adenine, 6 mM AMP, 200 mM EDTA) solutions and incubated for 3 min at 95°C to stop the reaction. The different radioactive compounds were separated by paper chromatography on Whatman 3MM strips using 28% NH$_4$OH, 50 mM EDTA as solvent for about 1 hr 30 min. Then, the substrates and products were visualized under a UV lamp at 254 nm and placed separately in vials in 2 ml of Scintran (VWR, Radnor, PA). The radioactivity in disintegrations per minute was measured in a Packard Tri-Carb 1600 TR liquid scintillation analyzer (PerkinElmer, Waltham, MA). The percentage of substrate transformation as a function of time was converted in nmol/min/mg protein and finally normalized to wild-type control values.

## Protein extraction and western blots

Thirty heads of 8–10-day-old males per genotype were homogenized in 30 µl Laemmli buffer containing protease inhibitor (cOmplete mini Protease Inhibitor Cocktail, Roche Diagnostics) using a Minilys apparatus (Bertin Instruments, Montigny-le-Bretonneux, France). The lysates were incubated on ice for 30 min and centrifuged at 8000 × g for 10 min at 4°C. The extracted proteins were heated at 95°C for 10 min. Western blots were performed as previously described (*Issa et al., 2018*). Briefly, proteins were separated in 4–12% Novex NuPAGE Bis-Tris precast polyacrylamide gels (Life Technologies) following the manufacturer's protocol in a MOPS-SDS running buffer. A semi-dry transfer was done onto polyvinylidene difluoride membranes (Amersham Hybond P 0.45 µm) using a Hoefer TE77 apparatus. Membranes were probed overnight at 4°C with mouse monoclonal anti-TH (1:5000, Cat# 22941, ImmunoStar) and mouse monoclonal anti-actin beta (1:5000, Cat# ab20272, Abcam, Cambridge, UK) that cross-reacts with *Drosophila* Actin 5C (Act5C). After incubation with horseradish peroxidase (HRP)-conjugated anti-mouse (1:5000, Cat# 115-035-146, Jackson ImmunoResearch) as secondary antibody, immunolabeled bands were revealed by chemiluminescence staining using ECL RevelBlOt Intense (Ozyme, Saint-Cyr-l'École, France, Cat# OZYB002-1000) and then digitally acquired with the ImageQuant TL software (GE Healthcare Life Science). Densitometry measures were made with the Fiji software and normalized to Act5C values as internal controls.

## Reverse transcription-PCR and quantitative PCR

Total RNA was isolated by standard procedure from 20 to 30 heads of 8-day-old males collected on ice and lysed in 600 µl QIAzol Reagent (QIAGEN, Venlo, Netherlands). 1 µg of total RNA was treated by DNase (DNase I, RNase-free, Thermo Fisher Scientific) according to the manufacturer's instructions. 5 µl of treated RNA was reverse transcribed using oligo d(T) primers (PrimeScriptRT Reagent Kit, Takara Bio). Then, at least 750 ng of the first-strand cDNA was amplified in 20 µl of reaction mixture using PrimeStar Max DNA polymerase (Takara Bio) with a Techne Prime Thermal Cycler apparatus (Bibby Scientific). The program cycles included 10 s denaturation at 98°C, 10 s annealing at 55°C, and 10 s elongation at 72°C, repeated 30 times. PCR product levels were measured after electrophoresis by densitometry analysis with the Fiji software. Data were normalized to amplification level of the ribosomal protein *rp49/RpL32* transcripts as internal control. Sequences of the primers used were for *HPRT*: forward 5'-GAGATACAAAATGGCGACCCGCAGCCCT, reverse, 5'-GCTCGGATCCTTATCATTACTAGGCTTTG (amplicon 686 bp); and for *rp49*: forward 5'- GACGCTTCAAGGGACAGTATC and reverse *rp49*, 5'AAACGCGGTTCTGCATGAG (amplicon 144 bp).

For RT-qPCR, approximately 40 ng of the first-strand cDNA was amplified in 10 µl of reaction mixture using the LightCycler 480 SYBR Green I Master reaction mix (Roche Applied Science, Mannheim, Germany) and a LightCycler 480 Instrument (Roche Applied Science). The program cycles included a 10 min preincubation step at 95°C, 40 cycles of amplification (10 s denaturation at 95°C, 10 s annealing at 55°C, 20 s elongation at 72°C), followed by a melting curves analysis for PCR product identification. Data were normalized to amplification level of the ribosomal protein *rp49/RpL32* transcripts as internal control. The genes analyzed and primer sequences used for qPCR are *AdoR*, forward 5'-GGAGAAATTGCGATCGGATGACAC, reverse 5'-TCTTCAGCGAACTCCGAGTGAATG; *Aprt*, forward 5'-AATCAGCGCGGAAGACAAGCTA, reverse, 5'-CCACCTTGCCGATGAGTTCA

GT; *DTH1*, forward 5'-GGATCGAAAGCCAACCAAGTG, reverse 5'-CTTGGGGACCAACTGCGCTTTA; *Ent2*, forward 5'-ACGGCAAGGGATCAACGTC, reverse 5'-CCGTGCAGCAGGAATATAAAGA; *rp49*, forward 5'-GACGCTTCAAGGGACAGTATC; reverse 5'-AAACGCGGTTCTGCATGAG.

## Adenosine assay

Adenosine was determined by ultra performance liquid cChromatography (UPLC). Here, 5 whole flies or 30 heads were homogenized in 120 µl 0.9% (w/v) NaCl using a Minilys apparatus (Bertin Instruments) and frozen at –80°C. After unfreezing and 5 min of microcentrifugation, 20 µl of 10% perchloric acid were added to 70 µl of the supernatant and the mixture was left for 5 min on ice. After a new centrifugation, 20 µl of a neutralization solution (made by mixing 3 volumes of 3 M $K_2CO_3$ with 1 volume of 6.4 mM NaOH containing 0.4 mg/ml bromothymol blue) were added and the mixture was centrifuged again before injection (5 µl). Samples were analyzed with a UV diode-array detector on an Acquity UPLC HSS T3 column (1,8 µm, 2.1 × 150 mm) (Waters Corporation, Milford, MA). The mobile phases consisted of Buffer A (30 mM ammonium acetate, pH 4.0 with 1:10,000 heptafluorobutyric acid [HPFA]) and Buffer B (acetonitrile with 1:10,000 HPFA) using a flow rate of 0.3 ml/min. Chromatographic conditions were 3.5 min 100% Buffer A, 16.5 min up to 6.3% Buffer B, 2 min up to 100% Buffer B, and 1 min 100% Buffer B. The gradient was then returned over 5 min to 100% Buffer A, restoring the initial conditions. Results were normalized to protein levels for each sample.

## Statistics

Statistical significance was determined using the Prism 6 software (GraphPad Software, La Jolla, CA). Survival curves for longevity experiments were analyzed using the log-rank test. Student's *t*-test was used to compare two genotypes or conditions, and one-way or two-way ANOVA with Tukey's, Dunnett's, or Sidak's *post hoc* multiple comparison tests for three or more conditions. Results are presented as mean ± SEM. Probability values in all figures: $*p < 0.05$, $**p < 0.01$, $***p < 0.001$.

## Acknowledgements

We are grateful to Pr. Hyder A Jinnah for the gift of HPRT cDNAs and to Dr Thomas Riemensperger for helpful discussions. Part of the phylogenetic analyses were performed using the computing facilities of the PRABI-AMSB bioinformatics platform, Laboratory of Biometry and Evolutionary Biology, Lyon, France. This work was supported by funding from CNRS and ESPCI Paris to SB and Association Malaury to ICP. CP is a recipient of PhD fellowships from the Association Lesch-Nyhan Action (LNA) and Labex MemoLife.

## Additional information

### Funding

| Funder | Grant reference number | Author |
| --- | --- | --- |
| Centre National de la Recherche Scientifique | Laboratory funding | Serge Birman |
| École Supérieure de Physique et de Chimie Industrielles de la Ville de Paris | Laboratory funding | Serge Birman |
| Association Malaury | Laboratory funding | Irène Ceballos-Picot |
| Association Lesch-Nyhan Action | Graduate Student Fellowship | Céline Petitgas |
| Labex MemoLife | Graduate Student Fellowship | Céline Petitgas |

The funders had no role in study design, data collection and interpretation, or the decision to submit the work for publication.

## Author contributions
Céline Petitgas, Conceptualization, Formal analysis, Validation, Investigation, Methodology, Writing – original draft, Writing – review and editing; Laurent Seugnet, Sandrine Marie, Conceptualization, Formal analysis, Investigation, Methodology; Amina Dulac, Rebecca Fima, Marion Strehaiano, Joana Dagorret, Formal analysis, Investigation; Giorgio Matassi, Formal analysis, Investigation, Methodology; Ali Mteyrek, Baya Chérif-Zahar, Methodology; Irène Ceballos-Picot, Conceptualization, Supervision, Funding acquisition, Validation, Writing – original draft, Writing – review and editing; Serge Birman, Conceptualization, Supervision, Funding acquisition, Validation, Writing – original draft, Project administration, Writing – review and editing

## Author ORCIDs
Laurent Seugnet (ID) http://orcid.org/0000-0003-1617-5721
Giorgio Matassi (ID) http://orcid.org/0000-0003-4923-226X
Serge Birman (ID) http://orcid.org/0000-0002-4278-454X

Reviewer #1 (Public review): https://doi.org/10.7554/eLife.88510.3.sa1
Reviewer #2 (Public review): https://doi.org/10.7554/eLife.88510.3.sa2
Reviewer #3 (Public review): https://doi.org/10.7554/eLife.88510.3.sa3
Author response https://doi.org/10.7554/eLife.88510.3.sa4

## Additional files

### Supplementary files
• MDAR checklist

### Data availability
All data generated or analysed during this study are included in the manuscript and supporting files; source data files have been provided for all the figures and linked figure supplements.

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
