## [Editor Report · eLife assessment]

The article looks at how dysregulated purine metabolism in mutants for the *Aprt* gene impacts survival, motor, and sleep behavior in the fruit fly. Interestingly, although several deficits arise from dopaminergic neurons, dopamine levels are increased in *Aprt* mutants. Instead, the biochemical change responsible for *Aprt* mutant neurobehavioral phenotypes appears to be a reduction in levels of adenosine. This **valuable** study suggests that *Drosophila Aprt* mutants may serve as a model for understanding Lesch–Nyhan disease (LND), caused by mutations in the human HPRT1 gene, and may also potentially serve as a model to screen for drugs for the neurobehavioral deficits observed in LND. The strength of evidence is **solid**.

---

## [Referee Report · Reviewer #1 (Public review)]

The current manuscript focusses on the adenine phosphoribosyltransferase (Aprt) and how the lack of its function affects nervous system function. It puts it into the context of Lesch-Nyhan disease, a rare hereditary disease linked to hypoxanthine-guanine phosphoribosyltransferase (HGPRT). Since HGPRT appears absent in *Drosophila*, the study focusses initially on Aprt and shows that aprt mutants have a decreased life-span and altered uric acid levels (the latter can be attenuated by allopurinol treatment). Moreover, aprt mutants show defects in locomotor reactivity behaviors. A comparable phenotype can be observed when specifically knocking down aprt in dopaminergic cells (in an adult-specific fashion). Interestingly, also glia-specific knock-down caused a similar behavioral defect, which could not be restored when re-expressing UAS-aprt, while neuronal re-expression did restore the mutant phenotype. Moreover, mutants, pan-neuronal and glia-specific RNAi for aprt caused sleep-defects. Based on immunostainings Dopamine levels are increased; UPLC shows that adenosine levels are reduced and PCR showed in increase of Ent2 levels are increased (but not AdoR). Moreover, aprt mutants display seizure-like behaviros, which can be partly restored by purine feeding (adenosine and N6-methyladenosine). Finally, expression of the human HGPRT also causes locomotor defects.

The authors provide a wide range of genetic experimental data to assess behavior and some molecular assessment on how the defects may emerge. It is clearly written, and the arguments follow the experimental evidence that is provided.

The findings provide a new example of how manipulating specific genes in the fruit fly allow the study of fundamental molecular processes that are linked to a human disease.

---

## [Referee Report · Reviewer #2 (Public review)]

The manuscript by Petitgas et al demonstrates that loss of function for the only enzyme responsible for the purine salvage pathway in fruit-flies reproduces the metabolic and neurologic phenotypes of human patients with Lesch-Nyhan disease (LND). LND is caused by mutations in the enzyme HGPRT, but this enzyme does not exist in fruit-flies, which instead only have Aprt for purine recycling. They demonstrate that mutants lacking the Aprt enzyme accumulate uric acid, which like in humans can be rescued by feeding flies allopurinol, and have decreased longevity, locomotion and sleep impairments and seizures, with striking resemblance to HGPRT loss of function in humans. They demonstrate that both loss of function throughout development or specifically in the adult ubiquitously or in all neurons, or dopaminergic neurons, mushroom body neurons or glia, can reproduce the phenotypes (although knock-down in glia does not affect sleep). They show that the phenotypes can be rescued by over-expressing a wild-type form of the Aprt gene in neurons. They identify a decrease in adenosine levels as the cause underlying these phenotypes, as adenosine is a neurotransmitter functioning via the purinergic adenosine receptor in neurons. In fact, feeding flies throughout development and in the adult with either adenosine or m6A could prevent seizures. They also demonstrate that loss of adenosine caused a secondary up-regulation of ENT nucleoside transporters and of dopamine levels, that could explain the phenotypes of decreased sleep and hyperactivity and night. Finally, they provide the remarkable finding that over-expression of the human mutant HGPRT gene but not its wild-type form in neurons impaired locomotion and induced seizures. This means that the human mutant enzyme does not simply lack enzymatic activity, but it is toxic to neurons in some gain-of-function form. Altogether, these are very important and fundamental findings that convincingly demonstrate the establishment of a *Drosophila* model for the scientific community to investigate LND, to carry out drug testing screens and find cures.

The authors have dealt with my concerns satisfactorily and have explained the instances in which resolving experimentally the criticisms raised would require a work effort well beyond the scope of a revision for this manuscript.

---

## [Referee Report · Reviewer #3 (Public review)]

The revised study provides better evidence to suggest that loss of Aprt activity in *Drosophila* provides a model for the loss of HGPRT activity in humans, which is causative for LND. Analysis of *Drosophila* Aprt mutations and RNAi-mediated knockdown reveals similar phenotypes to LND, particularly neurological defects, reduced nighttime sleep, and potentially seizures. LND is currently resistant to treatments and screening of a limited number of compounds in *Drosophila* has not identified a compound that can reduce all of the associated phenotypes. It is appropriate, therefore, that claims to have developed a clinically exploitable model for human LND have been toned down. Future drug screening may well prove profitable, but currently the evidence that *Drosophila* Aprt will be a suitable model for LND remains speculative.

The second approach adopted is to express a 'humanised mutated' form of HGPRT in *Drosophila*, which holds more promise for the development of a pharmacological screen. In particular, the locomotor defect is recapitulated but the seizure-like activity, whilst reported as being recapitulated, is debatable. A recovery time of 2.3 seconds is very much less than timings for typical seizure mutants. Nevertheless, the SING behaviour could be sufficient to screen against. However, this is not explored. With respect the short seizure duration, the authors cite similar findings for porin loss of function, but the cited study similarly did not employ anti-seizure drug exposure to validate that this phenotype is seizure related.

In summary, this is a largely descriptive study reporting the behavioural effects of an Aprt loss-of-function mutation. RNAi KD and rescue expression studies suggest that a mix of neuronal (particularly dopaminergic and possibly adenosinergic signalling pathways) and glia are involved in the behavioural phenotypes affecting locomotion, sleep and seizure. There remains insufficient evidence to have full confidence that the Arpt fly model will prove valuable for understanding / treating LND.

---

## [Author Response]

The following is the authors’ response to the original reviews.

**Reviewer #1 (Public Review):**
The current manuscript focuses on the adenine phosphoribosyltransferase (Aprt) and how the lack of its function affects nervous system function. It puts it into the context of Lesch-Nyhan disease, a rare hereditary disease linked to hypoxanthine-guanine phosphoribosyltransferase (HGPRT). Since HGPRT appears absent in *Drosophila*, the study focuses initially on Aprt and shows that aprt mutants have a decreased life-span and altered uric acid levels (the latter can be attenuated by allopurinol treatment). Moreover, aprt mutants show defects in locomotor reactivity behaviors. A comparable phenotype can be observed when specifically knocking down aprt in dopaminergic cells. Interestingly, also glia-specific knock-down caused a similar behavioral defect, which could not be restored when re-expressing UAS-aprt, while neuronal re-expression did restore the mutant phenotype. Moreover, mutants, pan-neuronal and pan-neuronal plus glia RNAi for aprt caused sleep-defects. Based on immunostainings Dopamine levels are increased; UPLC shows that adenosine levels are reduced and PCR showed in increase of Ent2 levels are increased (but not AdoR). Moreover, aprt mutants display seizure-like behaviors, which can be partly restored by purine feeding (adenosine and N6methyladenosine). Finally, expression of the human HGPRT also causes locomotor defects.The authors provide a wide range of genetic experimental data to assess behavior and some molecular assessment on how the defects may emerge. It is clearly written, and the arguments follow the experimental evidence that is provided. The findings provide a new example of how manipulating specific genes in the fruit fly allows the study of fundamental molecular processes that are linked to a human disease.

We thank the reviewer for his clear understanding and positive assessment of our work.

**Reviewer #2 (Public Review):**
The manuscript by Petitgas et al demonstrates that loss of function for the only enzyme responsible for the purine salvage pathway in fruit-flies reproduces the metabolic and neurologic phenotypes of human patients with Lesch-Nyhan disease (LND). LND is caused by mutations in the enzyme HGPRT, but this enzyme does not exist in fruit-flies, which instead only have Aprt for purine recycling. They demonstrate that mutants lacking the Aprt enzyme accumulate uric acid, which like in humans can be rescued by feeding flies allopurinol, and have decreased longevity, locomotion and sleep impairments and seizures, with striking resemblance to HGPRT loss of function in humans. They demonstrate that both loss of function throughout development or specifically in the adult ubiquitously or in all neurons, or dopaminergic neurons, mushroom body neurons or glia, can reproduce the phenotypes (although knock-down in glia does not affect sleep). They show that the phenotypes can be rescued by over-expressing a wild-type form of the Aprt gene in neurons. They identify a decrease in adenosine levels as the cause underlying these phenotypes, as adenosine is a neurotransmitter functioning via the purinergic adenosine receptor in neurons. In fact, feeding flies throughout development and in the adult with either adenosine or m6A could prevent seizures. They also demonstrate that loss of adenosine caused a secondary up-regulation of ENT nucleoside transporters and of dopamine levels, that could explain the phenotypes of decreased sleep and hyperactivity and night. Finally, they provide the remarkable finding that over-expression of the human mutant HGPRT gene but not its wild-type form in neurons impaired locomotion and induced seizures. This means that the human mutant enzyme does not simply lack enzymatic activity, but it is toxic to neurons in some gain-of-function form. Altogether, these are very important and fundamental findings that convincingly demonstrate the establishment of a *Drosophila* model for the scientific community to investigate LND, to carry out drug testing screens and find cures.

We thank the reviewer for his clear understanding and positive assessment of our work.

The experiments are conducted with great rigour, using appropriate and exhaustive controls, and on the whole the evidence does convincingly or compellingly support the claims. The exception is an instance when authors mention 'data not shown' and here data should either be provided, or claims removed: "feeding flies with adenosine or m6A did not rescue the SING phenotype of Aprt mutants (data not shown)". It is important to show these data (see below).

As recommended by the reviewer, these results are now shown in the new Figure S15.

Sleep is used to refer to lack of movement of flies to cross a beam for more than 5 minutes. However, lack of movement does not necessarily mean the flies are asleep, as they could be un-motivated to move (which could reflect abnormal dopamine levels) or engaged in incessant grooming instead.These differences are important for future investigation into the neural circuits affect by LND.

We agree that the method we used could overestimate sleep duration because flies that don't move do not necessarily sleep either, as it is the case with brain-dopamine deficient flies (Riemensperger et al., PNAS 2011). To address this issue, we have recorded video data showing that after 5 min of inactivity, wild-type and Aprt5 mutant flies are less sensitive to stimulation, indicating that they were indeed asleep. This is now shown in the new Figure S10 and mentioned on page 17, lines 338-339 in the main text. In addition, in this work we report that Aprt mutant flies have a nocturnal insomnia phenotype. Sleep overestimation is not, therefore, an issue that could challenge these results.

The authors claim that based on BLAST genome searchers, there are no HPRTI (encoding HGPRT) homologues in *Drosophila*. However, such a claim would require instead structure-based searches that take into account structural conservation despite high sequence divergence, as this may not be detected by regular BLAST.

To reinforce our conclusions about the lack of homologue of the human HPRT1 gene in *Drosophila*, we have now added a Results section about the evolution of HGPRT proteins on pages 6-7, lines 122150, and two phylogenetic analyses as new Figures S2 and S3 with more details in legends. We have also carried out structural similarity searches against the RCSB PDB repository. The structuralanalysis did not identify any relevant similarity with HGPRT 3D structures in Insecta (mentioned lines146-150). We hope these new analyses address the Reviewer's concerns. Furthermore, as shown in Table S2, no enzymatic HGPRT activity could be detected in extracts of wild-type *Drosophila*. A protein that would be structurally similar to human HGPRT but with a divergent sequence could not be involved in purine recycling without expressing HGPRT-like activity. In contrast, enzymatic Aprt activity could be easily detected in this organism (Figure S4 and Table S1).

This work raises important questions that still need resolving. For example, the link between uric acid accumulation, reduced adenosine levels, increased dopamine and behavioural neurologic consequences remain unresolved. It is important that they show that restoring uric acid levels does not rescue locomotion nor seizure phenotypes, as this means that this is not the cause of the neurologic phenotypes.

We agree with the reviewer about the potential importance of our results and the need to resolve the exact origin of the neurological phenotypes. This would need to be addressed in further studies in our opinion. The fact that allopurinol treatment did not improve the locomotor ability of Aprt5 mutant flies is now shown in Figure 1D, E to emphasize this result. Results showing that allopurinol does not rescue the bang-sensitivity phenotype of Aprt-deficient mutants are shown in Figure S14.

Instead, their data indicate adenosine deficiency is the cause. However, one weakness is that for the manipulations they test some behaviours but not all. The authors could attempt to improve the link between mechanism and behaviour by testing whether over-expression of Aprt in neurons or glia, throughout development or in the adult, and feeding with adenosine and m6A can rescue each of the behavioural phenotypes handled: lifespan, SING, sleep and seizures. The authors could also attempt to knock-down dopamine levels concomitantly with feeding with adenosine or m6A to see if this rescues the phenotypes of SING and sleep.

The reviewer is right. However, carrying out all these experiments properly with enough repeats will require about two more years of work. Because of that, they could not be included in the revision of the present article. Here we show that Aprt overexpression in neurons, but not in glia, rescues the SING phenotype of Aprt5 mutants (Figure 2B and 2E). We have also added in the revised article the new result that Aprt overexpression reduces transcript levels of DTH1, which codes for the neural form of the dopamine-synthesizing enzyme tyrosine hydroxylase (new Figure 5F).

Visualising the neural circuits that express the adenosine receptor could reveal why the deficit in adenosine can affect distinct behaviours differentially, and which neurologic phenotypes are primary and which secondary consequences of the mutations. This would allow them to carry out epistasis analysis by knocking-down AdoR in specific circuits, whilst at the same time feeding Aprt mutants with Adenosine.

Deciphering the specific circuits involved in the various effects of adenosine would indeed be extremely interesting. Unfortunately very few is currently known about the neural circuits that express AdoR in flies. No antibody is available to detect this receptor in situ and mutated AdoR gene coding for a tagged form of the receptor has not been engineered yet to our knowledge.

The revelation that the mutant form of human HGPRT has toxic effects is very intriguing and important and it invites the community to investigate this further into the future.To conclude, this is a fundamental piece of work that opens the opportunity for the broader scientific community to use *Drosophila* to investigate LND.

We sincerely thank the reviewer for his thoughtful and positive comments on our work.

**Reviewer #3 (Public Review):**
The study attempts to develop a *Drosophila* model for the human disease of LND. The issue here, and the main weakness of this study, is that *Drosophila* does not express the enzyme, HGPRT, which when mutated causes LND. The authors, instead, mutate the functionally-related *Drosophila* Aprt enzyme. However, it is unknown whether Aprt is also a structural homologue. Because of this, it will likely not be possible to identify pharmacological compounds that rescue HGPRT activity via a direct interaction (unless modelling predicts high conservation of substrate binding pocket between the two enzymes, etc).

As stated in our Provisional Responses prior to revision of the Reviewed Preprint, the enzymes APRT and HGPRT are actually known to be functionally and structurally related. We apologize for not providing this information in the original submission. This point is now made clearer in the revised article on page 39, lines 785-792. Indeed, both human APRT and HGPRT belong to the type I PRTases family identified by a conserved phosphoribosyl pyrophosphate (PRPP) binding motif, which is used as a substrate to transfer phosphoribosyl to purines. This binding motif is only found in PRTases from the nucleotide synthesis and salvage pathways (see: Sinha and Smith (2001) Curr Opin Struct Biol 11(6):733-9, doi: 10.1016/s0959-440x(01)00274-3). The purine substrates adenine, hypoxanthine and guanine share the same chemical skeleton and APRT can bind hypoxanthine, indicating that APRT and HGPRT also share similarities in their substrate binding sites (Ozeir et al. (2019) J Biol Chem. 294(32):11980-11991, doi: 10.1074/jbc.RA119.009087). Moreover, *Drosophila* Aprt and Human APRT are closely related as the amino acid sequences of APRT proteins have been highly conserved throughout evolution (see Figure S5B in our paper).

An additional weakness is that the study does not identify a molecule that may act as a lead compound for further development for treating LND. Rather, the various rescues reported are selective for only a subset of the disease-associated phenotypes. Thus, whilst informative, this first section of the study does not meet the study ambitions.

In this study, we identify adenosine and N6-methyladenosine as rescuers of the epileptic behavior in Aprt mutant flies (shown in Figure 7E, F). Interestingly, the same molecules have been found to rescue the viability of fibroblasts and neural stem cells derived from iPSCs of LND patients, in which de novo purine synthesis was prevented (discussed on page 38, lines 747-753). This suggests that the *Drosophila* model reported here could help to identify new genetic targets and pharmacological compounds capable to rescue HGPRT mutations in humans.

The second approach adopted is to express a 'humanised mutated' form of HGPRT in *Drosophila*, which holds more promise for the development of a pharmacological screen. In particular, the locomotor defect is recapitulated but the seizure-like activity, whilst reported as being recapitulated, is debatable. A recovery time of 2.3 seconds is very much less than timings for typical seizure mutants. Nevertheless, the SING behaviour could be sufficient to screen against. However, this is not explored.

We agree with the reviewer that it would be very interesting to do a pharmacological screen in this second LND model. However, we did not have the possibility to carry out such a screen yet.

In summary, this is a largely descriptive study reporting the behavioural effects of an Aprt loss-offunction mutation. RNAi KD and rescue expression studies suggest that a mix of neuronal (particularly dopaminergic and possibly adenosinergic signalling pathways) and glia are involved in the behavioural phenotypes affecting locomotion, sleep and seizure. There is insufficient evidence to have confidence that the Arpt fly model will prove valuable for understanding / treating LND.

Here we report many common phenotypes between the Aprt fly model and the symptoms of LND patients (reduced longevity, locomotor problems, sleep defects, overproduction of uric acid that is rescued by allopurinol treatment…). Moreover, APRT and HGPRT enzymes are both functional and structural homologues, as explained in our answers. We also found that the same drugs can rescue the seizure-like phenotype in Aprt-deficient flies and the viability of LND fibroblasts and neural stem cells, derived from iPSCs of LND patients, in which de novo purine synthesis is prevented (Figure 7E, F). In many respects, our results therefore suggest that Aprt mutant flies could be useful to better understand LND, and potentially to screen for new therapeutic compounds.

**From the Reviewing Editor:**
(1) How are the pathways of purine catabolism different between flies and mammals? How does the absence of HGPRT and presence of only AGPRT affect purine catabolism? When did HGPRT appear in evolution?

Purine catabolism is quite similar in flies and mammals, except for the lack of urate oxidase in primates, as described in Figure S1. We added words in the revised article about purine anabolism/catabolism pathways lines 123-126 (see below our detailed response to Reviewer 1’s Recommandations). HGPRT is present in Bacteria, Archea and Eukaryota, and nearly all animal phyla. However, BLAST search indicates that HGPRT homologues cannot be found in most insect species, such as *Drosophila*. To reinforce our conclusions about the lack of homologue of the human HPRT1 gene in *Drosophila melanogaster*, we have now added a Results section about the evolution of HGPRT proteins on pages 6-7, lines 122-150, and two phylogenetic analyses as new Figures S2 and S3 with details in legends.

In addition to BLAST a structural based modelling method should be used to establish the loss of HGPRT in *Drosophila*.

In agreement with the phylogenetic analyses, we have confirmed that no HGPRT enzymatic activity can be detected in wild-type *Drosophila* extract (Table S2). To complete these observations, as recommended by reviewer #2, we have carried out 3D structure-based searches in the RCSB Protein Data Bank. This enabled us to compare human HGPRT with all currently available protein structures. W found no *Drosophila* protein with a divergent sequence showing relevant structural similarity to human HGPRT. In contrast, this search identified proteins similar to human HGPRT in many other species of Eukaryota, Archea and Bacteria. This is now mentioned on page 7, lines 146-150 in the revised article.

(2) Of the three biochemical changes reported the change in dopamine levels should be validated by other methods given the unreliable nature of IHC.

As recommended by Reviewer #1, we have added the results of new experiments carried out by RTqPCR and Western blotting, which confirm the effect of Aprt mutation on brain dopamine levels. In addition, we added the consistent result that Aprt overexpression reduces transcript levels of DTH1. The results are shown in the new panels E to H of Figure 5 and mentioned in the text on page 20, lines 385-389.

(3) As suggested by reviewer 2 it would be helpful to clearly identify which of the three biochemical changes (DA, uric acid, adenosine) are responsible for the numerous behaviours tested. This is important because it is relevant for developing any therapeutic strategy arising from this study.

We agree that it would be very interesting to decipher the relationship between the different behaviors observed in mutant flies and the biochemical changes (dopamine, uric acid or adenosine). However, this would require a large amount of new experiments and it would probably double the size of our paper, which already includes many original data. In our opinion, such a detailed study should logically be the purpose of another article.

(4) There is concern regarding the robustness of the seizure data. Reviewer 3 has suggestions on how to address this.

See our answers to Reviewer 3’s recommendations below.

(5) Editorial corrections and changes suggested by reviewers 2 and 3 need to be addressed.

As indicated in our answers, we have taken into account and when possible addressed the corrections and changes suggested by the reviewers.

(6) It is recommended that the authors tone down the relevance of this model for LND, particularly in the abstract. The focus should be on stating what is actually delivered.

As recommended by the reviewing editor, and to take in account the reserved comments of reviewer #3, we have toned down our affirmation that our new fly models are relevant for LND in the last sentences of the Abstract and Discussion, and also added a question mark in the subtitle of the Discussion on line 777. As mentioned in our provisional responses to the Public Reviews, we would like to emphasize, however, that reviewers #1 and #2 expressed more confidence than reviewer #3 in the potential usefulness of our work. Reviewer #1 indeed stated that: “The findings provide a new example of how manipulating specific genes in the fruit fly allows the study of fundamental molecular processes that are linked to a human disease”, and reviewer #2 further wrote: "Altogether, these are very important and fundamental findings that convincingly demonstrate the establishment of a *Drosophila* model for the scientific community to investigate LND, to carry out drug testing screens and find cures”, and added: “To conclude, this is a fundamental piece of work that opens the opportunity for the broader scien2fic community to use *Drosophila* to inves2gate LND”.

**Reviewer #1 (Recommendations For The Authors):**
An important prerequisite for the current study is that there appears to be no HGPRT "activity" in *Drosophila*. It is initially stated that there was previously no "HGPRT activity observed" in two papers form the 70ies. It would be important to corroborate this notion and provide some background on the/catabolism pathways. How shared or divergent are these pathways between *Drosophila* and mammals?

In agreement with the pioneering studies of Becker (1974a, b), we have confirmed in this work that no HGPRT enzymatic activity can be detected in wild-type *Drosophila* extracts, as mentioned in Results on page 6, lines 127-130 and reported in Table S2. Purine catabolism is quite similar in flies and mammals, except for the lack of urate oxidase in primates, as shown in Figure S1. All the enzymes involved in purine anabolism/catabolim or recycling in humans have been conserved in *Drosophila* and humans, with the notorious exception of HPRT1.

If there is no HGPRT gene, but only the APRT ortholog, what would this mean for the metabolites?Our enzymatic assays on *Drosophila* extracts indicated that hypoxanthine and guanine cannot be recycled into IMP and GMP, respectively, contrary to adenine which can be converted into AMP in flies. In the absence of HGPRT activity, GMP and IMP could be produced by de novo purine synthesis, or, alternatively, synthesized from AMP, which can be converted into IMP by the enzyme AMPD, and then IMP can be converted into GMP by the enzymes IMPDH and GMPS. These metabolic pathways are depicted in Figure S1A.Is the lack of HGPRT specific for *Drosophila*, insects (generally in invertebrates)? I feel clarifying this would provide more insight into the motivation of the experimental approach.

As suggested by the Reviewer and the Reviewing Editor, we have addressed the evolution of HGPRT proteins more precisely in the revision. We have added a section on this subject in Results on pages 67, lines 122-150, and two phylogenetic analyses as Figures S2 and S3 with details in legends. A phylogenetic analysis was carried out a few years ago by Giorgio Matassi, who is now co-author of this paper. The most striking result was the great impact of horizontal gene transfer in the evolution of HGPRT in Insects (Figures S2 and S3). Our analysis of the phyletic distribution of HGPRT proteins revealed their striking rareness in Insecta, and in particular, their absence in Drosophilidae. The PSIBlast search detected however a significant hit in *Drosophila* immigrans (accession KAH8256851.1). Yet, this sequence is 100% identical to the HGPRT of the Gammaroteobacterium Serratia marcescens. Indeed, a phylogenetic analysis showed that *D. immigrans* HGPRT clusters with the Serratia genus (see Figure S3). This can be interpreted either a contamination of the sequenced sample, or as a very recent horizontal gene transfer event. The second scenario is more likely for the corresponding nucleotide sequences differ by 5 synonymous substitutions (out of 534 positions). A powerful approach to try to understand the "origin" of the *D. immigrans* protein would be to analyze whether horizontal gene transfer has affected its chromosomal neighbours. This approach, proposed previously by G. Matassi (BMC Evol Biol, 2017, 17:2, doi: 10.1186/s12862-016-0850-6), is highly demanding in terms of computing time and would require an ad hoc study. We hope that these new analyses address the Reviewer's concerns.

On the mechanistic side on how the behavioral defects may arise, the authors show that dopaminergic neurons (and glia cells) are involved. One interesting finding is that dopamine immunostainings suggest increased dopamine levels. However, immunostainings are notorious for artifacts and do not provide a strong quantitative assessment. I feel it would be helpful to have an alternative technique to corroborate this finding.

We agree with the reviewer and we added the results of further confirmatory experiments in the four new panels E-H of Figure 5, showing that: (1) the transcript levels of DTH1 (encoding the neuronal isoform of the dopamine-synthesizing enzyme tyrosine hydroxylase in *Drosophila*) are increased in Aprt5 mutants compared to wild-type flies (new Figure 5E), (2) consistent with this, DTH1 transcript levels were found in contrast to be decreased when Aprt was overexpressed ubiquitously in flies (new Figure 5F), (3) Western blot experiments showed that DTH1 protein levels are also increased in Aprt5 mutant flies compared to controls (new Figure 5G-H).

**Reviewer #2 (Recommendations For The Authors):**
As mentioned in the public review, the behavioural phenotypes of decreased lifespan, SING, sleep and seizures could be tested for all manipulations: feeding with allopurinol, adenosine and m6A, and combining this with knock-down dopamine levels in PAMs or MBs. This could help dissect the relationship between mutations in Aprt and behaviour.

We thank the reviewer for these suggestions, and, indeed, we would have liked to do all these experiments. However, as mentioned in our responses to the Public Reviews, carrying out these experiments properly with sufficient repeats would require about two more years of work. We have already accumulated a large amount of data, so we have decided to publish our results at this stage in order to make our new fly models available to the scientific community. We are giving careful and due consideration to these experimental proposals and we hope to continue our investigation on this topic in the future.

It would also be helpful to find out which neurons and glia express AdoR. Perhaps there are already tools available the authors could test or at least check with the scRNAseq Fly Atlas (public Scope database).

Following the reviewer’s recommendation, we have checked the scRNAseq Fly Atlas for AdoR expression in the brain, compared to that of ple (encoding tyrosine hydroxylase) and Eaat1 (encoding the astrocytic glutamate transporter). As shown in the image below, the results are not very informative. AdoR appears to be expressed in rather widespread subsets of neurons and glial cells, that partly overlap with ple and Eaat1 expression. Further work would be required to identify more precisely the neurons and glial cells expressing AdoR in the brain.

**Author response image 1. sa4fig1:** 

Page 7, line 161: use of the word 'normalize'. "We tried to normalise uric acid content in flies..." would best to use 'rescue' instead, as normalisation in science has a different meaning.

We modified this word as suggested.

Page 9 line 203: 'genomic deficiencies that cover': the genetic term is 'uncover', as a deficiency for a locus reveals a phenotypes, thus it is said 'a gene uncovered by xx deficiency".

Thank you for this helpful remark. We corrected this in line 221.

Page 10, lines 206-208: 'allopurinol treatment did not improve the locomotor activity...". These are important observations that should be best presented within the main manuscript Figure 1.

As recommended, we have transferred the graphs of Figure S5 to new panels D and E of Figure 1.

Figure 4: please indicate genotypes in the figure, where no information is given that these are UASAprt-RNAi experiments.

We added the complete genotype in Figure 4G, and also in Figure S12C and D. Thank you for noting that.

Page 25 line 491: "None of these drugs was able to rescue the SING defects (data not shown)". Either provide the data or remove this claim.

We have added these data in the new Figure S15.

Statistical analyses: details are provided in the methods, but the name of test and multiple comparisons corrections should be also provided in the legends.

Thank you very much for the careful proofreading. This was an oversight and we have added the information in all legends of the revised article.

**Reviewer #3 (Recommendations For The Authors):**
This is a difficult manuscript to appreciate. The abstract and introduction suggest that the study is to identify novel treatments for a human disease (LND) by development of a *Drosophila* model. Much of the results, however, are focussed to describing the consequences to purine metabolism of the Aprt mutation. To my mind, a rewrite to focus on the latter would be beneficial. The potential applicability to LND would be best restricted to the discussion.

We apologize for not making our goals clearer. Our purpose was to find out if purine recycling deficiency could lead to metabolic and neurobehavioral disturbances in *Drosophila*, as it is the case in human LND patients when HGPRT is mutated. Interestingly, we observed that mutation of the only purine recycling enzyme in flies, Aprt, did induce defects in part comparable to that of LND in humans, including overproduction of uric acid that is rescued by allopurinol treatment, reduced longevity, and various neurobehavioral phenotypes including bang-sensitive seizure, sleep defects and locomotor impairments. We also identified adenosine and N6-methyladenosine as rescuers of the epileptic behavior in these mutants. These drugs were also identified as therapeutic candidates in screens based on iPSCs from LND patients. This suggests that Aprt deficiency in Drosophila could be used as a model to better understand this disease and find new therapeutic targets.

Regardless of the above comment, the concluding sentence of the abstract is inappropriate. This study does not show that *Drosophila* can be used to identify a cure for LND.

We agree with the Reviewer that the last sentence of the abstract was too affimative. As also recommended by the reviewing editor, we have modified this sentence in the abstract and other sentences in the text in order to tone down the affirmation that our new fly models are relevant for LND. See our answers to the Reviewing Editor above for details.

Indeed, I would challenge the premise that screening against a functional, but unknown if structural, homologue (Aprt) will ever provide an exploitable opportunity. To meet this statement, this study needs to identify a treatment that rescues all of the behavioural phenotypes associated with the Aprt mutation, in addition to rescuing the influences of the mis-expression of mutated HGPRT.

APRT and HGPRT are both functionally and structurally related. Both human APRT and HGPRT belong to the type I PRTases family identified by a conserved phosphoribosyl pyrophosphate (PRPP) binding motif, which is used as a substrate to transfer phosphoribosyl to purines. This binding motif is only found in PRTases from the nucleotide synthesis and salvage pathways (see: Sinha and Smith (2001) Curr Opin Struct Biol 11(6):733-9733-9, doi: 10.1016/s0959-440x(01)00274-3). The purine substrates adenine, hypoxanthine and guanine share the same chemical skeleton and APRT can bind hypoxanthine, indicating that APRT and HGPRT also share similarities in their substrate binding sites (Ozeir et al. (2019) J Biol Chem. 294(32): 11980-11991, doi: 10.1074/jbc.RA119.009087). This point has been made clearer in the Discussion page 39, in lines 785-792.. Finally, *Drosophila* Aprt and Human APRT are closely related as the amino acid sequences of APRTs have been highly conserved throughout evolution (shown in Figure S5B).

With respect to expression of the mutated HGPRT: the short seizure recovery time of 2.3 seconds is not very convincing evidence of a seizure phenotype. This is far below the timings reported for typical BS mutations. Because of this, the authors should run a positive control (e.g. one of the wellestablished BS mutations: parabss, eas or jus) to validate their assay. Moreover, was the seizure induced by the Aprt mutation (17.3 secs - again a low value) rescued by prior exposure to an antiepileptic? Could this behaviour be, instead, related to the SING locomotor phenotype?

The assay we used to test for bang-sensitivity has been validated in previous articles from different laboratories. We agree that the recovery times we observed were shorter than those of the BS mutations mentioned by the reviewer. However, we could cite another *Drosophila* BS mutant, porin, that shows similarly short recovery times (2.5 and 6 sec, according to the porin alleles tested, Graham et al. J Biol Chem. 2010, doi: 10.1074/jbc.M109.080317). This is now mentioned on page 36 lines 717-720. In addition, the BS phenotype we observed with Aprt mutants was robust and highly significant compared to control flies (Figure 7). We did not try to rescue this phenotype by exposing the flies to an antiepileptic, but we do not think that it can be related to the SING phenotype. Indeed, providing adenosine or N6-methyladenosine to Aprt5 mutant flies was able to rescue the BS phenotype (Figure 7E, F), but did not rescue the locomotor defects (new Figure S15). Moreover, SING performances of Aprt5 mutant flies at 8 or 30 d a. E. are decreased nearly in almost identical way (Figure 1C), while we observed an effect on BS behavior at 30 d a. E., which implies that the SING and BS behaviors are most likely unrelated.

Line 731 states that 'Aprt mutants show a typical BS phenotype' - whilst accurate to some extent (e.g. the behaviour depicted in the supp videos), it should be made clear, it should be made clear that the recovery time is uncharacteristically short and thus differs from typical BS mutations.

We have corrected the sentence in the revised article to mention that (page 36, lines 717-718).

Line 732 stating that BS phenotype is often linked to neuronal activity - what other links would there be? Even if via glia or other tissues the final effect is via neurons.

We have modified this sentence (page 36, line 720).

The introduction and, particularly, the discussion are overly long and, in the case of the latter, repetitive of the results text. Pruning to make the paper more concise would be very beneficial. Removal of the extensive speculation about how DA and adenosine may interact would help in this regard (line 688 onwards). Indeed, in many places the discussion morphs into a review.

We agree with the reviewer on this point, and have therefore done our best to shorten theIntroduction and Discussion, which are now 24% and 21% shorter, respectively, in the revised article compared to the original submission.

The applicability of using *Drosophila* Aprt mutations to screen for compounds that may treat LND is predicated on some degree of similarity in either enzyme structure or metabolic pathways. A discussion of how relevant, therefore, studying Aprt is needs to be included. Given the authors insights - where should potential new rugs be targeted to?

As stated above, we now mention in the article that APRT and HGPRT share similarities in their structure. In addition, the metabolic pathways between humans and *Drosophila* have been largely conserved (shown in Figure S1B).